# Global transboundary synergies and trade-offs among Sustainable Development Goals from an integrated sustainability perspective

Huijuan Xiao [1,2], Sheng Bao [3], Jingzheng Ren [2,4,5] ✉, Zhenci Xu [6,7] ✉, Song Xue [2] & Jianguo Liu [8] ✉

Domestic attempts to advance the Sustainable Development Goals (SDGs) in a country can have synergistic and/or trade-off effects on the advancement of SDGs in other countries. Transboundary SDG interactions can be delivered through various transmission channels (e.g., trade, river flow, ocean currents, and air flow). This study quantified the transboundary interactions through these channels between 768 pairs of SDG indicators. The results showed that although high income countries only comprised 14.18% of the global population, they contributed considerably to total SDG interactions worldwide (60.60%). Transboundary synergistic effects via international trade were 14.94% more pronounced with trade partners outside their immediate geographic vicinity than with neighbouring ones. Conversely, nature-caused flows (including river flow, ocean currents, and air flow) resulted in 39.29% stronger transboundary synergistic effects among neighboring countries compared to non-neighboring ones. To facilitate the achievement of SDGs worldwide, it is essential to enhance collaboration among countries and leverage transboundary synergies.

All United Nation (UN) member states have implemented 17 Sustainable Development Goals (SDGs) in pursuit of peace and prosperity for all people and the planet[1,2]. The three main pillars of sustainability—economy, society, and environment—encompassed these goals. Sustainability is often approached from two perspectives: weak and strong sustainability[3]. Weak sustainability posits that each of these pillars holds equal weight and that the pillars are interchangeable[3,4]. Strong sustainability prioritises the environmental pillar[3,4]. However, recent research has introduced a new perspective on sustainable development: integrated sustainability[4–6]. This concept extends

beyond traditional weak and strong sustainability perspectives and incorporates the spillover effects generated by the transboundary interactions across regions as a fourth pillar, alongside the original three pillars[6]. These spillover effects represent the interplay of the three original pillars of sustainable development between two or more regions[5,7]. In the current interconnected world, transboundary interactions across countries may positively or negatively affect SDGs in various other countries[8]. Global sustainable development cannot be achieved by countries that act alone. Communication between countries can promote interdisciplinary programs and multilateral

[1]Department of Civil and Environmental Engineering, The Hong Kong University of Science and Technology, Hong Kong SAR, China. [2]Department of Industrial and Systems Engineering, The Hong Kong Polytechnic University, Hong Kong SAR, China. [3]Otto Poon C. F. Smart Cities Research Institute, Department of Land Surveying and Geo-Informatics, The Hong Kong Polytechnic University, Hong Kong SAR, China. [4]Research Center for Resources Engineering Towards Carbon Neutrality, The Hong Kong Polytechnic University, Hong Kong SAR, China. [5]Department of Industrial and Systems Engineering, Research Institute for Advanced Manufacturing, The Hong Kong Polytechnic University, Hong Kong SAR, China. [6]Department of Geography, The University of Hong Kong, Hong Kong SAR, China. [7]Shenzhen Institute of Research and Innovation, The University of Hong Kong, Hong Kong SAR, China. [8]Center for Systems Integration and Sustainability, Department of Fisheries and Wildlife, Michigan State University, East Lansing, MI, USA. ✉e-mail: jzhren@polyu.edu.hk; xuzhenci@hku.hk; liuji@msu.edu

collaborations, help policymakers formulate coherent plans and strategies, and effectively unlock transboundary SDG interaction potential[9].

Widespread interactions exist between SDGs across country borders[1,8,10–13], such as technological spillovers from multinational corporations and profits from international trade. These transboundary synergies may help receiving countries achieve their SDGs[14]. Conversely, there may also be transboundary trade-offs, such as wastewater flow into transboundary rivers, which may hamper the achievement of SDGs in countries receiving wastewater[15–17]. However, little attention has been paid to determining the impacts of these transboundary interactions on SDGs[18]. Quantifying transboundary SDG interactions is challenging because countries are connected through different transmission channels, and the outcomes of transboundary SDG interactions can vary[19–21]. Regarding human-caused flows, a common channel of transboundary SDG interactions is international trade[22], which may have environmental and socioeconomic impacts on trade partners owing to the water[23,24], carbon[25], and labour used to produce goods and embodied in trade[26] (Fig. 1a). Additionally, nature-caused flows connect many countries (Fig. 1b). For instance, under uncontrolled pollution conditions, air and wind may transport airborne pollutants into neighbouring countries[27] or even to distant countries through intercontinental transport[21,28,29], thereby compromising air quality and human health in receiving countries[28]. Furthermore, pollutants discarded in waterways affect local communities and neighbouring countries.

This study investigates integrated sustainability in the context of the SDGs to determine whether each SDG indicator exerts positive or negative effects on the others and to quantify these effects. A conceptual framework incorporating different channels of transboundary SDG interactions was first proposed. This framework was built based on metacoupling (e.g., human–nature interactions within and between neighbouring and distant countries)[30], as shown in Fig. 1. This study classified these channels into two broad categories, each of which interacted with the performance of the SDGs of other countries in different ways: human-caused flows (e.g., international trade) and nature-caused flows (e.g., river flow, ocean currents, and air flow) (Fig. 1). Second, this study examined 768 pairs of SDG indicators to evaluate how an individual SDG indicator of a country interacts with other countries' indicators through different channels[31–33]. The pairs of indicators were identified as having causal relationships (e.g., energy intensity and $CO_2$ emission intensity indicators as the interaction generator and receiver, respectively), which can be derived from a database about SDG interactions[8] (Supplementary information Tables S1 and S2). Finally, this study proposed a spatial interaction index to quantify the overall magnitude of transboundary interactions between the performance of the SDGs in one country and those in other countries. This index can be divided into transboundary synergistic and trade-off effects and is a scorecard (score: 0–100) used to indicate the magnitude of transboundary interactions. Based on available data for 2010 to 2020, 121 countries were chosen for analysis (Supplementary information Table S3). The findings of this study can aid in improving the understanding, monitoring, and careful management of transboundary SDG interactions.

## Results
### Transboundary SDG interaction linkages across countries
Through the transmission channels of both international trade and nature-caused flows (incorporating river flow, ocean currents, and air flow), the transboundary synergistic linkages were more pronounced than their trade-off counterparts. Specifically, amongst the transboundary linkages, which include synergistic and trade-off linkages, 73.68% of the linkages resulting from international trade were synergistic (Fig. 2a, b). Similarly, 81.82% of linkages originating form nature-caused flows were synergistic (Fig. 2c, d). These results also highlight that, compared with interaction linkages resulting from nature-caused flows, linkages originating from international trade were generally more susceptible to counterproductive effects, potentially undermining joint efforts towards the SDGs. To provide further clarity, within the sphere of international trade, trade-off linkages accounted for 26.32%

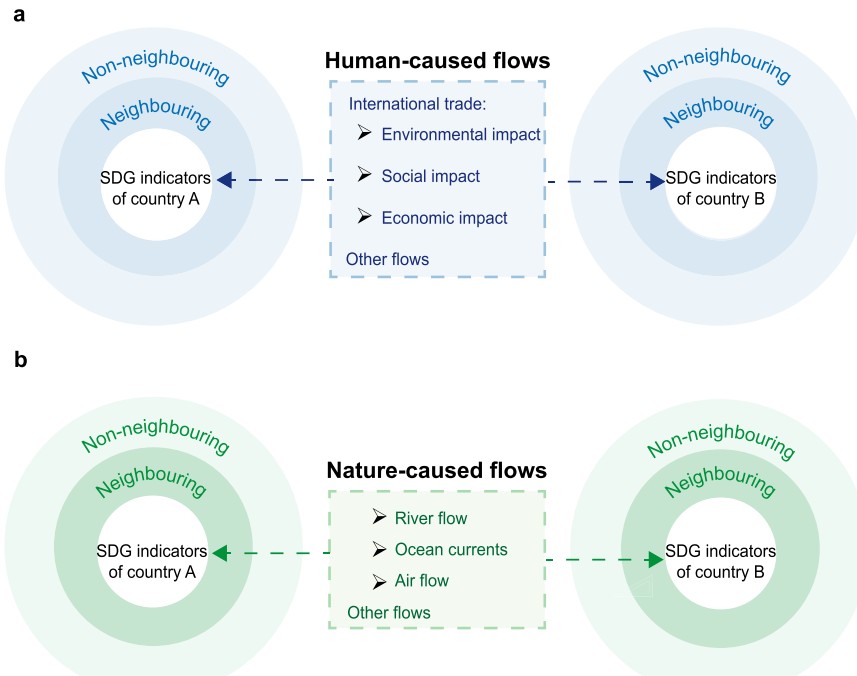

**Fig. 1 | Conceptual framework of transboundary interactions of Sustainable Development Goals (SDGs) across countries. a** Human-caused flows. **b** Nature-caused flows. Two categories of channels can create SDG interactions between countries: human-caused flows and nature-caused flows. The receiving countries of the interactions are classified as either neighbouring or non-neighbouring countries.

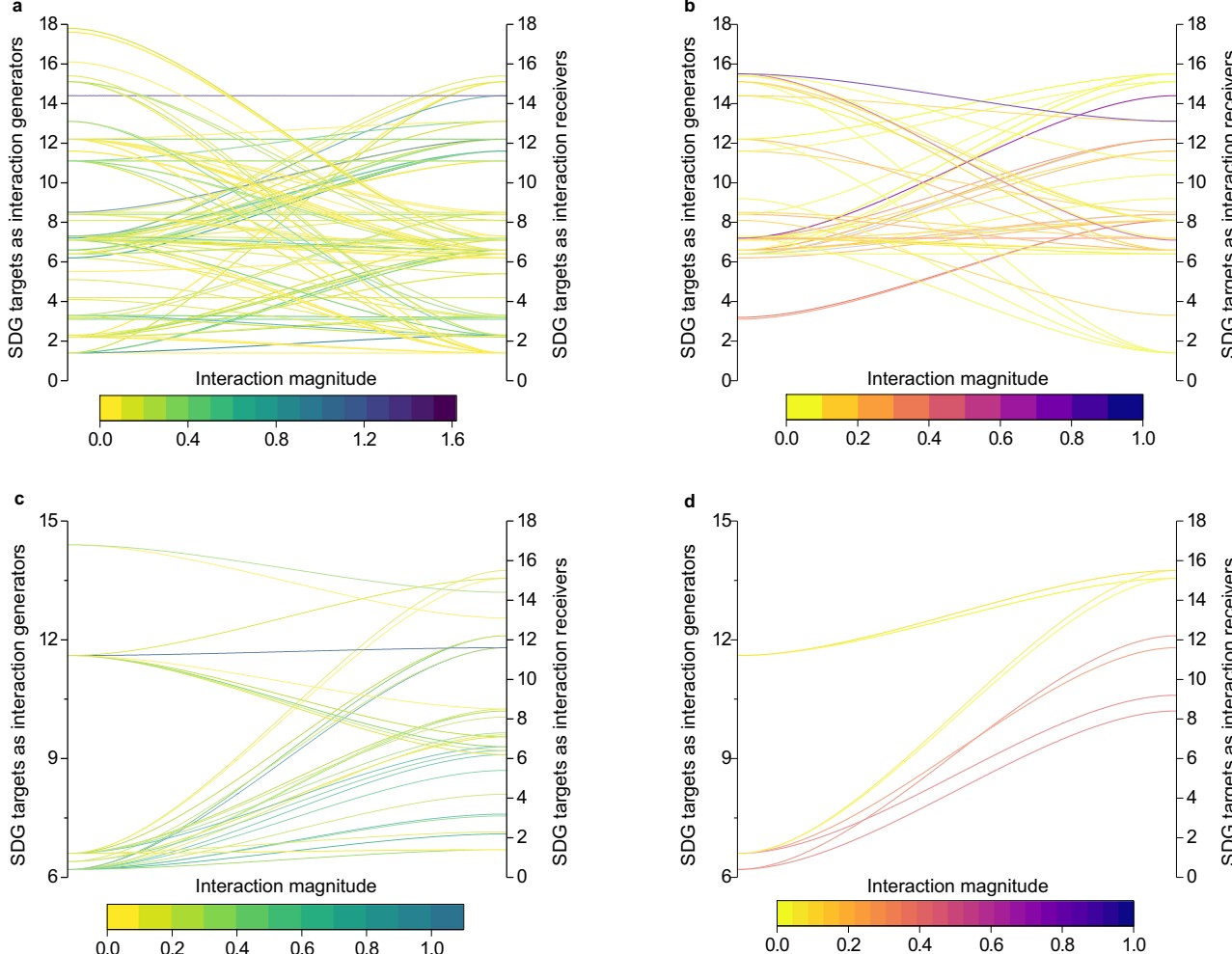

**Fig. 2 | Transboundary synergistic and trade-off linkages across SDG indicators among countries. a** Transboundary synergistic linkage across SDG indicators via international trade. **b** Transboundary trade-off linkage across SDG indicators via international trade. **c** Transboundary synergistic linkages across SDG indicators through nature-caused flows. **d** Transboundary trade-off linkages across SDG indicators through nature-caused flows. For better demonstration, the SDG indicators belonging to the same SDG target were grouped together based on the UN Global Indicator Framework for Sustainable Development Goals developed by the Inter-Agency and Expert Group on SDG Indicators (IAEG-SDGs). The left and right axes respectively denote the SDG targets belonging to SDG 1 to SDG 17, with the former serving as interaction generators and the latter as interaction receivers. The colour bars show the absolute values of spatial coefficients that are significant, which serve to represent the magnitude of transboundary interactions. These values were derived from 768 regression models based on spatial econometric methods, elaborated further in the Methods section. A darker colour, corresponding to a higher absolute value of the spatial coefficient, signifies stronger transboundary interactions.

(calculated as 100−73.68%) of the total SDG interaction linkages (Fig. 2a, b). This percentage is notably higher than the 18.18% (calculated as 100−81.82%) associated with nature-caused flows, as shown in Fig. 2c, d.

In the international trade channel, indicators related to target 7.1 (to ensure universal access to affordable, reliable, and modern energy services) had the most (29) linkages with the SDG indicators in other countries (Fig. 2a, b). These indicators were linked to various basic human needs and the environment in other countries, such as basic drinking water and sanitation services (four linkages with target 1.4), agricultural productivity (two linkages with target 2.3), water-use efficiency (two linkages with target 6.4), housing (three linkages with target 11.1), and biodiversity (two linkages with target 15.5) (Fig. 2a, b). For instance, via the channel of international trade, the spatial lag term of target 7.1 proved to be both significant and positive in Table 1. This implies that the achievement of target 6.4 in certain countries could be promoted by synergistic effects stemming from the progress their trade partners have made towards target 7.1. This extensive network of linkages may be primarily attributed to the fundamental role of energy in many sectors. The production, distribution, and consumption of energy through international trade can have far-reaching transboundary impacts on various aspects of society and the environment.

In the channel of nature-caused flows, the SDG indicator that affected the most SDG indicators in other countries was related to target 6.6. (protect and restore water-related ecosystems) (Fig. 2c, d). Attempts to improve the performance of target 6.6 in interconnected countries may interact with the performance of 19 SDG indicators in focal countries (Fig. 2c, d). For example, the spatial lag of term of target 6.6 (river flow) is 0.094, with significance at the 1% level (Table 1), suggesting the progress of SDG 1.4 of some countries can be promoted by the other countries' actions towards achieving target 6.6 through the transboundary rivers. The actions of other countries focused on protecting and restoring water-related ecosystems may have transboundary SDG impacts, thereby creating numerous benefits for focal countries. These benefits include promoting equitable access to basic water and sanitation services (four linkages with target 1.4), contributing to the decoupling economic growth from environmental degradation (two linkages with target 8.4), and sustainable management and efficient use of natural resources (three

**Table 1 | Empirical results of transboundary interactions based on a two-stage instrumental variable (2SIV) estimation of the spatial econometric model**

| First stage estimation | | Explained variable: target 6.4 | | Explained variable: target 1.4 | |
|---|---|---|---|---|---|
| GDP | 0.091*** (0.033) | Spatial lag of target 7.1 (trade flow) | 0.189*** (0.043) | Spatial lag of target 6.6 (trade flow) | 0.038*** (0.013) |
| Population | 0.071** (0.035) | Time lag | 0.234*** (0.079) | Spatial lag of target 6.6 (river flow) | 0.094*** (0.013) |
| Governance | 0.034 (0.030) | Economy | 0.020 (0.026) | Time lag | −0.003 (0.059) |
| Internet | 0.028 (0.034) | Education | −0.029 (0.028) | Economy | −0.001 (0.007) |
| Export value | 0.102*** (0.035) | Technology | 0.053** (0.024) | Environment | 0.017 (0.010) |
| Technology | 0.038 (0.027) | Governance | 0.045 (0.028) | Education | 0.019* (0.010) |
| | | Agriculture | −0.006 (0.025) | Governance | 0.018** (0.009) |
| | | Residual from the first stage | 0.039* (0.023) | Population | −0.002 (0.009) |
| | | | | Residual from the first stage | −0.018** (0.007) |

The indicators chosen to represent SDG targets 7.1, 6.4, 1.4, and 6.6 are proportion of population with primary reliance on clean fuels and technology, water use efficiency, proportion of population using basic sanitation services, and lakes and rivers seasonal water area (% of total land area), respectively. Supplementary information Tables S5 and S6 show the detailed information regarding the additional variables and the rationale behind their selection. Standard errors are provided in parentheses. Significance at the 1%, 5% and 10% levels is denoted by ***, ** and *, respectively.

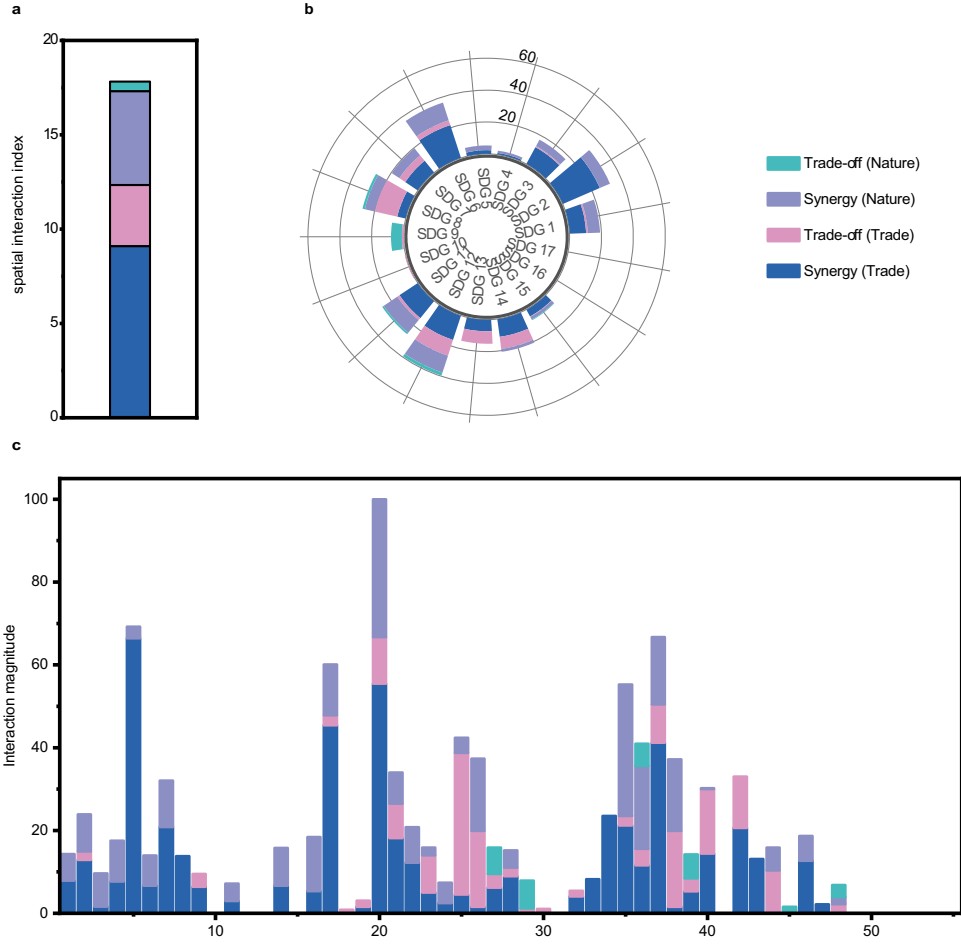

**Fig. 3 | Spatial interaction index and its components. a** Spatial interaction index of countries worldwide. **b** Interaction magnitude of 17 SDGs. **c** Interaction magnitude of 55 SDG indicators. Both the spatial interaction index and interaction magnitude are divided into four components: synergistic effects via nature-caused flows, trade-off effects via nature-caused flows, synergistic effects via international trade, and trade-off effects via international trade. The numbers displayed on the horizontal axis of **c** represent the identifiers for the 55 SDG indicators included in this study (Supplementary information Table S1).

linkages with target 12.2) (Fig. 2c, d). Furthermore, these actions can promote the conservation, restoration, and sustainable use of terrestrial and inland freshwater ecosystems and their services in focal countries (two linkages with target 15.1), as well as reduce the degradation of natural habitats and biodiversity loss (two linkages with target 15.5) (Fig. 2c, d). Fig. 2 also reveals that a single SDG indicator in other countries can influence both its counterpart and

various other SDG indicators in focal countries. For example, air pollutants (target 11.6) in other countries, through air flow, can profoundly impact both ambient air quality and 17 other indicators across multiple SDGs in focal countries (Fig. 2c, d). This ripple effect may influence health outcomes and economies—leading, for example, to a potential reduction in work time and productivity, which aligns with SDG 8 (promoting decent work and economic growth)

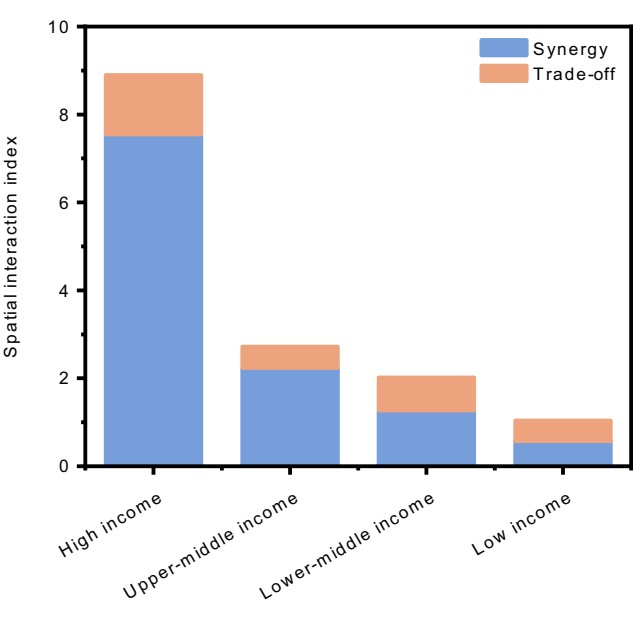

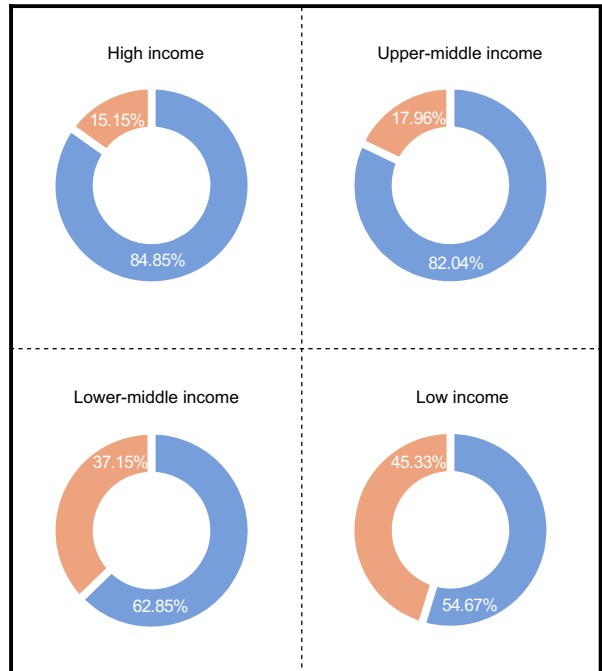

**Fig. 4 | Magnitude and components of transboundary SDG interactions by income group. a** Spatial interaction index by income groups. **b** Share of the components of spatial interaction index by income groups. The sum of synergistic effects and trade-off effects equals the spatial interaction index. There are four income groups: high, upper-middle, lower-middle, and low income groups (Supplementary information Table S4). The blue in the pie charts represents the proportion of transboundary synergistic effects.

(Fig. 2c, d). Moreover, it can also impact biodiversity, as represented by SDG 15 (life on land), by modifying habitats and harming wildlife (Fig. 2c, d). This discovery underscores the complex and interconnected nature of SDG interactions mediated by natural flows.

## Magnitude of transboundary SDG interactions of countries worldwide

A spatial interaction index with a scale of 0 to 100 was devised to quantify the overall magnitude of transboundary SDG interactions. This index includes both synergistic and trade-off effects and considers transmission channels via both human-caused and nature-caused flows. A higher index indicates more substantial transboundary SDG interactions with other countries, and the index consists of four components, as shown in Fig. 3a. The aggregate of two components, namely, the transboundary synergistic effects via nature-caused flows and transboundary synergistic effects via international trade, denotes the total magnitude of the transboundary synergistic effects. For the countries worldwide, this magnitude represented 78.97% of the spatial interaction index (Fig. 3a) and was 3.76 times stronger than the total magnitude of the transboundary trade-off effects. This finding indicates that transboundary SDG interactions between countries can facilitate SDG accomplishment. Compared to the transmission channel via nature-caused flows, transboundary synergistic effects via international trade had a lower share among the countries worldwide. Specifically, the contribution of synergistic effects via nature-caused flows to the total transboundary interactions via nature-caused flows of the countries worldwide reached 90.71%, whereas the contribution of synergistic effects via international trade to the total transboundary interactions via international trade was 73.76% (Fig. 3a). Among the 17 SDGs, SDG 12 (responsible consumption and production) showed the most potent transboundary synergistic effects, scoring 39.38, followed by SDG 2 (zero hunger) and SDG 6 (clean water and sanitation), which

scored 32.13 and 36.49, respectively (Fig. 3b). Among the 55 SDG indicators, target 6.6 (protect and restore water-related ecosystems) received the strongest transboundary synergistic effects (88.74), equivalent to the sum of transboundary synergistic effects through international trade (55.45) and nature-caused flows (33.29) (Fig. 3c). Considering the net effects of transboundary SDG interactions (transboundary synergistic effects minus transboundary trade-off effects), target 6.6 (protect and restore water-related ecosystems) again ranked first, demonstrating the strongest net effects (77.49) owing to its large transboundary synergistic effects (88.74) and small transboundary trade-off effects (11.25). This finding suggests that advancements in SDG indicators in other countries can significantly advance target 6.6 in the focal countries (Fig. 3c).

## Magnitude of transboundary SDG interactions by income group

This study divided 121 countries into four groups based on the World Bank country classification (2022–2023). Compared to low, lower-middle, and upper-middle income groups, high income countries bear a greater responsibility for the influence of their domestic actions on the achievement of the 17 SDGs in other countries, as the magnitude of their transboundary SDG interactions accounted for the largest proportion of the total transboundary SDG interactions of the four income groups (sum of spatial interaction index), at 60.60% (Fig. 4a). High income countries demonstrated strong transboundary interactions with other countries; however, the population of high income countries over 2010–2020 accounted for an average of only 14.18% of the global population, based on data from the World Bank. Despite representing a relatively small fraction of the global population, high income countries are often characterised by robust economies, advanced technologies, and considerable political influence, which may amplify their roles in SDG interactions. Furthermore, when analysing the components of transboundary SDG interactions, the transboundary synergistic effects/trade-off effects of all high income

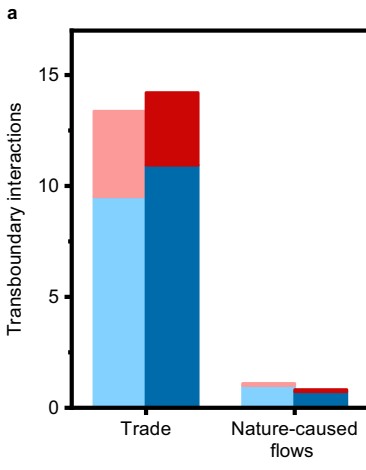

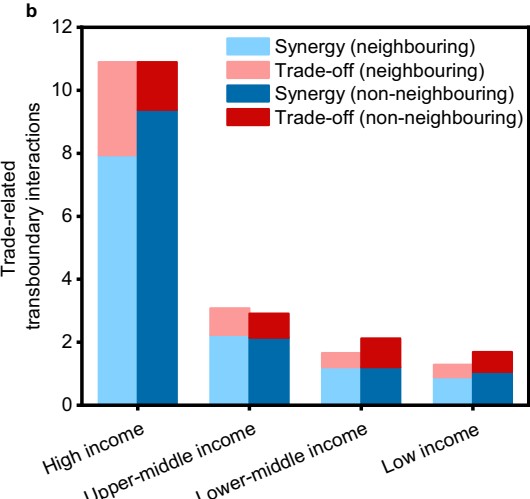

**Fig. 5 | Comparison of the transboundary SDG interactions with neighbouring and non-neighbouring countries. a** Magnitude of transboundary interactions with neighbouring and non-neighbouring countries. **b** Comparisons of the magnitude of transboundary SDG interactions among high, upper-middle, lower-middle, and low income countries. The blue and red represent the transboundary synergistic effects and transboundary trade-off effects, respectively. The light blue and dark blue colours respectively indicate the synergistic effects with neighbouring and non-neighbouring countries. In contrast, the light red and dark red indicate the trade-off effects with neighbouring and non-neighbouring countries, respectively.

countries constituted a large percentage of those globally, at 64.95% and 44.06%, respectively (Fig. 4a). High income countries showed the largest share of transboundary synergistic effects in their total transboundary SDG interactions (84.85%) compared to the other three income groups (Fig. 4b), suggesting that transboundary SDG interactions generated by high income countries may considerably promote the achievement of SDGs in their connected countries.

## Transboundary SDG interactions with neighbouring and non-neighbouring countries

The intensity of the transboundary effects of the countries worldwide differed depending on geographic proximity. Non-neighbouring countries derived more benefits from transboundary interactions facilitated by international trade; however, neighbouring countries derived greater benefits from transboundary interactions via channels of nature-caused flows (Fig. 5a). Specifically, the transboundary synergistic effects were 14.94% more pronounced in interactions between trade partners that did not share borders compared with their neighbouring counterparts (Fig. 5a). Conversely, through the transmission channel of nature-caused flows, neighbouring countries showed transboundary synergistic effects that were 39.29% stronger than those observed between non-neighbouring countries (Fig. 5a). Transboundary trade-off effects were 17.81% stronger between neighbouring trade partners than between non-neighbouring partners (Fig. 5a). Through the nature-caused flow channel, non-neighbouring countries exhibited transboundary trade-off effects that were 1.88% stronger than those between neighbouring countries (Fig. 5a). The net effects (i.e., synergistic effects minus trade-off effects) between non-neighbouring countries through international trade were 35.46% stronger than those between neighbouring nations (Fig. 5a). Contrarily, for the transmission channel of nature-caused flows, the net effects between neighbouring countries were 45.59% more robust than those between non-neighbouring countries (Fig. 5a).

An analysis of trade relationships revealed that all four income groups—high, upper-middle, lower-middle, and low income—demonstrated stronger synergistic effects than trade-off effects with both neighbouring and non-neighbouring trade partners. The share of synergistic effects among the total interactions (combining synergistic and trade-off effects) varied by income group and by whether the trading partners were neighbours. Specifically, the shares for high,

upper-middle, lower-middle, and low income countries with their neighbouring partners were 72.83%, 72.46%, 72.96%, and 69.07%, respectively (Fig. 5b). In contrast, their shares with non-neighbouring partners were 86.07%, 73.61%, 56.76%, and 62.73%, respectively (Fig. 5b). Interestingly, high income countries tended to establish notably more intense synergistic relationships with trade partners outside their immediate geographic vicinity than with neighbouring trade partners (Fig. 5b). Specifically, high income countries showed 18.14% stronger synergistic effects with non-neighbouring trade partners than with neighbouring ones (Fig. 5b). This can be attributed to the extensive global practices and international influence of high income countries. High income countries often have widespread networks of investments and trade relationships worldwide, facilitating stronger interactions with non-neighbouring countries. Participation in various international accords and organisations encourages these countries to extend their relationships beyond their immediate geographic sphere, fostering more intensive interactions globally. Moreover, their relatively advanced technological infrastructure enables efficient communication and transportation over long distances.

## Discussion

This study quantifies transboundary interactions among 121 countries in relation to 768 SDG indicator pairs from 2010 to 2020. This assessment was conducted through various channels, including international trade, river flow, ocean currents, and air flow, by employing an integrated sustainability perspective[4]. This study makes a key contribution by quantifying the magnitude and direction of the fourth pillar of integrated sustainability: the spillover effects caused by human−nature interactions. Therefore, sustainable development, as considered in this study, rests on four pillars: (1) social, (2) environmental, (3) economic, and (4) spillover effects[4]. These pillars correlate with four key principles: (1) people, (2) planets, (3) prosperity, and (4) peace and partnership[4,6].

Some research credits globalisation and openness with benefiting sustainability and economic development by invoking Ricardo's theory of comparative advantage[5,6,34]. However, other scholars argue that openness and globalisation run contrary to sustainability goals, based on the pollution haven hypothesis[6,35,36]. While theories of comparative advantage highlight the welfare gains of interconnectedness, the pollution haven hypothesis introduces an important caveat regarding

potential cross-border regulatory distortions and their impact on sustainability outcomes in integrated economies at different development levels. The study demonstrated that transboundary synergistic effects through international trade were the dominant form of interactions across borders, revealing the overall positive impact of globalisation and openness on advancing global sustainability in an interconnected world. While trade-offs exist in some issues, the predominance of cross-border coordination benefits underscores how collective progress is enhanced through continued cooperation at a global scale. Addressing a single SDG indicator within one country can automatically strengthen not only the same indicator but other indicators across connected nations through transboundary synergistic effects. Conversely, if these transboundary synergies develop in negative ways, such as a country failing to progress on certain indicators or regressing, they may result in vicious cycles in which setbacks are multiplied and transmitted to other countries[8]. This highlights the risks and emphasizes the necessity to convert vicious inter-country cycles into virtuous ones. Systemic interlinkages form either virtuous or vicious cycles, indicating that transformations must be pursued intentionally to initiate desirable co-benefits and multiplication effects across borders. Pursuing progress in a coordinated manner across countries could set the stage for mutually reinforcing advances in the SDGs at a global scale.

Tobler's first law of geography—that everything is related to everything else, but near things are more related than distant ones— serves as a foundational principle in numerous research fields, including spatial analysis in epidemiology[37], crime pattern analysis[38], economic development[39,40], and environmental issues[41]. The continuing relevance and applicability of this law are evidenced by the broad range of methodologies and concepts that have been developed based on it. In modern times, as sustainability issues increasingly intersect with geographical considerations, there has been a increasing interest in revisiting and further exploring Tobler's law, especially with respect to SDGs[42]. While the law's core principle remains valid in many instances, studies have illuminated the complex ways in which distance can shape interrelations, sometimes counterintuitively[16]. For instance, some studies have highlighted habitat losses triggered by distant consumption through international trade[43,44]. The negative effects of distant activities on sustainable fisheries further challenge this geographical principle[45,46]. The results of this study suggested that the synergistic effects were 14.94% more pronounced in interactions between trade partners that did not share borders compared with those between neighbouring counterparts. Due to globalisation, different countries have become more connected and less geographically limited through international trade. Globally, non-neighbouring countries can benefit from comparative advantages by diversifying their traded goods and services, allowing them to interact more with each other than with neighbouring countries. While Tobler's law remains valuable for understanding geographical influences, this research has revealed the importance of considering a broader array of factors, including non-proximate influences.

Transboundary SDG interactions are global issues that transcend individual nations. Beyond traditional place-based governance approaches with a focus on a country's territory, it is significant to adopt a flow-based perspective. This considers each country in the context of its associations with others by identifying, monitoring, and managing areas where key flows originate, progress between borders, and ultimately terminate[5,6,47]. This study advocates that countries collaborate to find solutions through international organisations that act as bridges to facilitate global policymaking and support the achievement of the 2030 Agenda. Some international organisations (e.g., the UN and World Trade Organization) were formed to implement adequate measures to address global issues. Two main measures are proposed to discourage transboundary trade-off effects and encourage synergistic effects. One is to internalise the costs and benefits of

transboundary SDG interactions. Countries that generate transboundary trade-off effects could be asked to provide adequate compensation, discouraging activities that impose a cost on an unrelated third party. Countries that generate transboundary synergistic effects can internalise these benefits through subsidisation, which could incentivise them to increase synergistic effects. This study evaluates the magnitude and components of transboundary SDG interactions, providing a foundation for countries worldwide to consider actions that inadvertently generate trade-offs in other countries, while reaping the benefits of synergies. The other measure is to establish a globally tradable pollution permit that presents countries with legally acceptable pollution limits. A tradable permit system has the substantial advantage of allowing efficient exchange, which helps maintain the overall level of pollution by allowing one potential polluter to purchase permits from another. A well-known example of this trading system is the Emission Trading System of the European Union, established in 2005. Thus, addressing transboundary SDG interactions requires more effective transboundary solutions and multilateral governance to achieve global sustainability.

In addition to international trade, river flow, ocean currents, and air flow, other cross-border exchanges shape SDG interactions[48,49]. For example, owing to the high-volume nature of seaborne freight, maritime shipping is well-suited for transporting goods across international borders in regions with extensive coastlines[50]. Cargo vessels can accommodate the bulk shipping of diverse goods across long distances in a relatively efficient manner compared with other forms of international transport[50]. This strengthens economic cooperation and trade opportunities between coastal countries[50]. Future studies should further explore the impacts of various human-caused transboundary flows, such as maritime transportation[50], technology transfer, investment, knowledge sharing, human migration, disease dissemination, and information diffusion, once data become available[51]. The influence of nature-caused flows, including animal migration, seed dispersal, and disease spread, is also worth investigating[52]. Capturing these additional linkages may provide deeper insights into the complex interconnected relationships between countries' progress towards achieving the SDGs. Modelling flows such as ocean currents and wind patterns poses interesting methodological challenges given their multidirectional, changing dynamics over varying temporal and spatial scales. However, greater precision in characterising connectivity tendencies may considerably enhance our understanding of sustainability linkages. Future research should investigate novel approaches to systematically tracking variations in flow vectors—such as harnessing remote sensing data—and integrating this directional flow of information into spatial regression frameworks. This may entail simulating transport processes or calibrating networks via hydrodynamic or atmospheric modelling. Capturing the full complexity of flow regimes may provide unprecedented insights into the causal relationships among different countries.

## Methods

### SDG indicator selection and data sources

There are 17 SDGs with 169 targets and 231 unique indicators within the global indicator framework[1]. This study chose the years 2010–2020 and 121 countries for analysis based on the best available data. A list of the 121 countries was included in this study (Supplementary information Table S3). This study included 55 indicators constructed using robust data and applied to a broad range of countries (Supplementary information Table S1). These 55 indicators were selected from the Indicators and Monitoring Framework for the Sustainable Development Goals developed by the UN Sustainable Development Solutions Network, the UN Global Indicator Framework for Sustainable Development Goals developed by the IAEG-SDGs, and some published studies[53,54]. The values of these indicators ranged from 0 (worst performance) to 100 (best performance)[2,54].

## Transboundary SDG interactions across countries

This study proposed four interconnected steps to evaluate how SDG indicators interact with indicators across countries through different transmission channels.

## Step 1: Match SDG indicator pairs with causal relationships

This study first determined whether there was a causal relationship between the SDG indicator pairs. Based on the 55 SDG indicators, 768 SDG indicator pairs showing potential causal relationships were identified, derived from an interactive repository of SDG interactions[8] (Supplementary information Table S2). By conducting a systematic literature review (65 global scientific assessments and UN flagship reports and 112 relevant scientific articles), this interactive repository recorded causal relationships across SDG targets covering the 17 SDGs[8]. Based on the following steps, this study further quantitatively verified whether the 768 SDG indicator pairs exhibited interactions between different countries through a variety of channels.

## Step 2: Construct spatial weight matrices for different channels of transboundary SDG interactions

Many channels connect two or more countries. This study divided these channels into human-caused and nature-caused flows (Fig. 1). This study used a spatial weight matrix ($W$) with diagonal elements equal to zero to represent each channel, as shown in the following equation:

$$W = [w_{ij}]_{N \times N} = \begin{bmatrix} 0 & w_{12} & \cdots & w_{1N} \\ w_{21} & 0 & \cdots & w_{2N} \\ \vdots & \vdots & 0 & \vdots \\ w_{N1} & w_{N2} & \cdots & 0 \end{bmatrix}, i \text{ and } j = 1, 2, 3, \ldots, N \quad (1)$$

where $w_{ij}$ indicates the weight between country $i$ and country $j$. $N$ represents the total number of countries. The channels of transboundary SDG interactions with respect to international trade in year $t$ were constructed based on multiregional input–output (MRIO) tables ($W_{trade,t} = [w_{ijt}]_{N \times N}$), and MRIO tables from years 2010 to 2020 were obtained from the Eora26 database[55,56]. Several other MRIO databases exist, including EXIOBase3, WIOD, and GTAP; however, this study chose Eora26 because it has a higher country coverage (189 countries) and can provide the most up-to-date MRIO tables[55,56]. The value-added of international trade between one country and another is indicated in each cell of the trade weight matrix (unit: USD).

This study divided the transboundary nature-caused flows into three categories: river flow, ocean currents, and air flow, as follows:

(1) River flow channel ($W_{river} = [w_{ij}]_{N \times N}$): When a transboundary river crosses these countries, some water-related SDG indicators from different countries can be linked through river flow. SDG 6 relates to clean water and sanitation; therefore, 102 indicator pairs related to SDG 6 were assumed to be connected via river flow channels (Supplementary information Table S2). Global geographical information of the rivers was derived from the HydroSHEDS database at a resolution of 15 arc-seconds[57,58]. The database displays over eight million river reaches worldwide, with more than 120,000 being the most downstream reaches of connected river basins[57,58]. These were used to identify the entire river network belonging to this basin and determine which countries share the entire river network[57,58]. The results revealed 2126 transboundary rivers worldwide. Subsequently, a weight matrix was constructed, with each cell indicating the aggregation of river flow between the two countries. To calculate this, this study added the average long-term discharge estimates of all river reaches between the two countries, which were obtained from the HydroSHEDS database[57,58]. The unit of the matrix was cubic meters per second. Greater river flow between the two countries indicates a stronger connection.

(2) Ocean current channel ($W_{maritime} = [w_{ij}]_{N \times N}$): Some SDG indicators in one country may be influenced by some ocean-related SDG indicators in countries with which they share sea areas. SDG 14 is related to the preservation and sustainable exploitation of oceans, seas, and their resources to foster sustainable development; thus, the 45 indicator pairs related to SDG 14 in coastal countries were assumed to be linked via ocean current channels (Supplementary information Table S2). Only coastal countries were included in the evaluation of indicators under SDG 14 (life below water). To construct a weight matrix for ocean currents across coastal countries, this study first excluded inland countries based on the Central Intelligence Agency World Factbook. Of the 121 countries, 91 were coastal and 30 were inland (Supplementary information Table S7). Then, based on maritime boundaries, this study identified each coastal country's neighbouring countries. The UN Conventions on the Law of the Sea defines maritime boundaries as territorial waters and contiguous and exclusive economic zones. Each matrix cell was filled with 1 or 0, indicating whether or not the two countries were linked by ocean currents.

(3) Air flow channel ($W_{air} = [w_{ij}]_{N \times N}$): Some air-related indicators (target 11.6: fine particulate matter) from various countries can influence certain SDG indicators of a particular country due to the movement of air (Supplementary information Table S1). In this study, a spatial weight matrix based on the inverse distance was constructed to represent the air flow connections between countries. This study calculated the distance between 121 countries based on their centroids. The transboundary SDG interactions of 36 indicator pairs related to the target 11.6 were estimated using this spatial weight matrix (Supplementary information Table S2).

This study is grounded in the metacoupling framework, an integrated conceptual construct examining the human–nature interplay within a coupled human–nature system, adjacent to that system and from distant locations[16,30,52]. This framework encompasses all flow types relevant to human and natural systems. Determination of the spatial weight matrix is guided by four key criteria: (1) Relevance: The chosen transmission channel should mirror the real-world transmission mechanism of the SDG indicator. For example, water-related SDG indicators may be interlinked through transboundary rivers. Consequently, the spatial weight matrix, represented by river flows across countries, was utilised to examine the transboundary interactions of indicators related to SDG 6 (clean water and sanitation). (2) Timeliness: Transmission channels influenced by socio-economic conditions, such as international trade, are dynamic and frequently change over time. Consequently, data series must be updated regularly, published promptly, and be made available for the most recent years to accurately reflect these changes. (3) Coverage: The data must adequately define the relationships between any two countries included in the study. They should provide a comprehensive understanding of the interactions and connections between these countries, offering a broad scope that does not neglect critical relationships. (4) Data availability and quality: The transmission channel data must represent the most accurate measure of a specific issue. They should be obtained from reliable national or international sources to ensure credibility and reliability. Considering these selection criteria, this study incorporated different flow types that exist across countries: trade flows (human-caused flows), river flows, ocean currents, and air flows (nature-caused flows).

## Step 3: Construct row-standardised spatial weight matrices

Row standardisation suggests that each spatial weight in a matrix is divided by its row sum, as shown below:

$$w_{ij}^s = \frac{w_{ij}}{\sum_{j=1}^{N} w_{ij}} \quad (2)$$

where $w_{ij}$ and $w_{ij}^s$ indicate the weight between country $i$ and country $j$, and the weight after row standardisation, respectively.

## Step 4: Quantify transboundary SDG interactions

Spatial econometric models were used to explore the transboundary SDG interactions of SDG indicator pairs under different transmission channels. Endogeneity issues may arise in statistical analyses when an explanatory variable is correlated with an error term, leading to biased and inconsistent estimates. These complexities require specific techniques to ensure accurate and reliable results[59–63]. Different techniques can be used to address the endogeneity problem[59–63]. To deal with the endogeneity issues caused by the elements of the spatial weight matrix involving socioeconomic indicators[64–68], this study applied the 2SIV estimation method based on the control function method to explore the spatial spillover effects of SDG indicators in a panel dataset[64]. The first stage was estimated using the following regression model:[64]

$$\ln F_{it} = \eta_i + \mathbf{X_{1it}}\boldsymbol{\gamma} + \varepsilon_{it}, t = 1, 2, \dots, T \tag{3}$$

$F_{it}$ denotes the trade flow in country $i$ at period $t$. $\boldsymbol{\gamma}$ is a vector of the coefficients of explanatory variables. $\mathbf{X_{1it}}$ is a list variables measuring the economy, population, government effectiveness, access to the internet, performance of export sectors, and technological level of country $i$[69] (Supplementary information Table S5). Based on the residual $\widehat{\varepsilon_{it}}$ from the first-step estimation, this study considered the following model for the second-stage estimation:[64]

$$\ln SDG_{imt} = c_i + \rho_1 \ln SDG_{im,t-1} + \lambda_1 \sum_{j \neq i} w^1_{ijt} \ln SDG_{jnt} + \mathbf{X_{2it}}\boldsymbol{\beta} + \delta\hat{\varepsilon}_{it} + v_{it} \tag{4}$$

$SDG_{imt}$ and $SDG_{jnt}$ are a pair of SDG indicators that were determined in Step 1, which respectively indicate the interaction receiver $m$ and generator $n$. $\rho_1$ is the scale coefficient. $\lambda_1$ is the spatial coefficient which can be used to measure the transboundary SDG interactions under the transmission channel of international trade. $w^1_{ijt}$ is a spatial weight matrix related to international trade between country $i$ and country $j$ in year $t$. $\boldsymbol{\beta}$ is a vector of coefficient of the explanatory variables. $\mathbf{X_{2it}}$ denotes the explanatory variables (Supplementary information Table S6). The positive variables were transformed by taking their natural logarithms in the spatial dynamic panel data model.

Multiple transmission channels could operate simultaneously[70–73]. Indicators related to SDG 6 (clean water and sanitation), SDG 14 (life below water), and SDG target 11.6 (fine particulate matter) could be affected both through international trade and nature-caused flows (transboundary river flow, ocean currents, and air flow). This study employed higher-order spatial econometric models to account for real-world complexity. These models can incorporate more than one spatial weight matrix and, thus, characterise various types of spatial dependence. Spatial weight matrix $w^2_{ij}$ was specifically utilised to represent channels related to nature-caused flows. $\lambda_2$ is a spatial coefficient used to evaluate transboundary SDG interactions under the transmission channel of nature-caused flows.

$$\ln SDG_{imt} = c_i + \rho_1 \ln SDG_{im,t-1} + \lambda_1 \sum_{j \neq i} w^1_{ijt} \ln SDG_{jnt} + \lambda_2 \sum_{j \neq i} w^2_{ij} \ln SDG_{jnt}$$
$$+ \mathbf{X_{2it}}\boldsymbol{\beta} + \delta\hat{\varepsilon}_{it} + v_{it} \tag{5}$$

## Creating the spatial interaction index and its decomposition

Domestic actions aimed at achieving SDGs may result in transboundary interactions with other countries. This study proposed a spatial interaction index to quantify the overall magnitude of transboundary interactions across all transmission channels. As the spatial coefficients (both $\lambda_1$ and $\lambda_2$) of the same explained variable (SDG indicator) from Step 4 were comparable, this study summed the absolute values of any coefficients (both positive and negative coefficients) that were significant at the 10% level or above. This provided a total impact measure of transboundary interactions on a specific SDG indicator.

Subsequently, min–max normalisation was performed on each total impact value to determine the interaction magnitude of each SDG indicator. This standardisation process scaled the values to a uniform range of 0 to 100. Bringing comparable transboundary interactions onto a unified scale allowed for an easy assessment of the relative influence across different SDG indicators. Finally, the arithmetic average of all the standardised values was calculated to derive the overall spatial interaction index. Ranging from 0 to 100, a higher spatial interaction index implied stronger transboundary interactions between countries.

The spatial interaction index can be divided into four distinct components: synergistic effects through nature-caused flows, trade-off effects through nature-caused flows, synergistic effects via human-caused flows, and trade-off effects via human-caused flows. First, the interaction magnitudes of the four components for each SDG indicator were determined. This was accomplished by multiplying the interaction magnitude of each SDG indicator (on a scale of 0–100) by the respective percentage shares of these components. We obtained these percentage shares from the absolute values of coefficients that were significant and calculated them as a proportion of the total sum. Subsequently, the components of the spatial interaction index were obtained by computing the arithmetic average of the interaction magnitudes for each component across all SDG indicators. This approach allows for a more nuanced understanding of the different factors contributing to transboundary SDG interactions.

Using SDG indicator 7.1.1 (access to electricity) as an example, this study analysed the magnitude and direction of impacts on a country's performance in achieving indicator 7.1.1 from progress on SDG indicators in other countries. In Step 1, it was identified that the achievement of indicator 7.1.1 could be influenced by indicator 6.4.1 (water use efficiency) and indicator 7.3.1 (energy intensity). Spatial weight matrices were constructed to represent connections between countries through river basins and trade networks. Subsequently, this study row-standardised the weight matrices. This normalisation process prepared the data for spatial econometric modelling. The last step involved using spatial econometric models to calculate the spatial coefficients. These coefficients (both $\lambda_1$ and $\lambda_2$) indicated the direction and magnitude of transboundary interactions on indicator 7.1.1 outcomes in the focal country, respectively. If both coefficients were significant at least 10%, the study would sum their absolute values and standardised this total into a single index from 0 to 100. This example shows how the performance of indicator 7.1.1 in focal countries could be influenced by other countries' progress on indicator 6.4.1, through shared river flows powering hydropower, and indicator 7.3.1, through energy used in internationally traded goods and services.

## Transboundary interaction by income group

This study divided 121 countries into four groups based on the World Bank country classification by income level (2022–2023) (Supplementary information Table S4). Income level was measured as gross national income per capita in current USD values. Subsequently, this study compared the transboundary interactions exerted by each income group. This was achieved by constructing an updated spatial weight matrix. In this revised matrix, each row retained the value of countries belonging to a specific income group, while all other values were set to zero. Following this, Steps 3 and 4 were replicated by employing the proposed spatial interaction index to contrast the magnitude of SDG interactions across different income brackets. In this analysis, the focus was strategically directed toward the trade-related transboundary interactions of these four income groups rather than interactions facilitated by nature-caused flows. The unique nature of the sparse spatial weight matrix of ocean currents and river flows presents an intriguing challenge for quantifying transboundary interactions using spatial econometric models, which has opened new avenues for future exploration.

### Transboundary SDG interactions with neighbouring and non-neighbouring countries

Transboundary SDG interactions between neighbouring and non-neighbouring countries may vary[16,47]. In this study, neighbouring countries refer to countries with a common vertex, land boundary, or maritime boundary, whereas non-neighbouring countries indicate countries without any common vertex, land boundary, or maritime boundary[16]. $W_{neighbour} = [w_{ij}]_{M \times M}$ is a spatial weight matrix related to neighbouring countries. Each element of $W_{neighbour}$ was filled with 1 or 0, indicating whether or not the two countries were neighbours. For the common vertex or land boundary, neighbours were identified based on a queen-contiguity-based spatial weight matrix ($W_{queen} = [w_{ij}]_{M \times M}$). For the common marine boundary, neighbours were identified based on $W_{marine} = [w_{ij}]_{M \times M}$. The spatial weight matrices related to non-neighbouring countries can be obtained after excluding the neighbouring countries of each country. This study then repeated Steps 3 and 4 and used the proposed spatial interaction index to compare the SDG interaction magnitude in non-neighbouring countries to that in neighbouring countries.

It is essential to acknowledge that the definitions of neighbouring and distant regions are subject to contextual variations, and there is no universally recognised measure that unequivocally designates a region as neighbouring or distant[74]. In certain research endeavours that seek to investigate the impacts of channels closely related to distance, distant regions can be identified through the application of various distance thresholds, thereby facilitating a more comprehensive understanding of the subject matter. To gain further insight, future studies should compare the impacts of transboundary interactions using different distance thresholds to distinguish between neighbouring and distant regions.

## Data availability

The data generated in this study are available in the Supplementary Information. Multiregional input-output (MRIO) tables can be obtained from the Eora26 database (https://worldmrio.com/eora26/). The geographical information of rivers globally can be derived from the HydroSHEDS database (https://www.hydrosheds.org/). The data of SDG indicators can be collected from SDG Global Database by UN (https://unstats.un.org/sdgs/dataportal), World Bank (https://www.worldbank.org/), and Our World in Data (https://ourworldindata.org/).

## Code availability

All computer code used in conducting the analyses summarized in this paper is available from the corresponding author upon request. The code for the spatial econometric models can be accessed in the paper with the https://doi.org/10.1111/ectj.12069.

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

## Acknowledgements
The work described in this paper was supported by a grant from Research Institute for Advanced Manufacturing (RIAM), The Hong Kong Polytechnic University (Project No. 1-CD4J, Project ID: P0041367) (J.R.), a grant from Research Centre for Resources Engineering towards Carbon Neutrality (RCRE), The Hong Kong Polytechnic University (PolyU) (Project No.1-BBEC, Project ID: P0043023) (J.R.), a grant from Research Grants Council of the Hong Kong Special Administrative Region, China-General Research Fund (Project ID: P0042030, Funding Body Ref. No: 15304222, Project No. B-Q97U) (J.R.), U.S. National Science Foundation (Grants No. 1924111, 2033507 and 2118329), Michigan AgBioResearch (J.L.), National Natural Science Foundation of China (grant #42101249), and the University of Hong Kong HKU-100 Scholars Fund (Z.X.).

## Author contributions
J.L., Z.X., and J.R. designed and supervised the study. H.X. performed the analysis and prepared the manuscript. H.X. and S.B. established the model and compiled the code. H.X. and S.X. collected the data. J.L., Z.X., J.R., and S.X. reviewed and revised the manuscript.

## Competing interests
The authors declare no competing interests.
