## [Peer Review File · Nature Communications]

Global transboundary synergies and trade-offs among Sustainable Development Goals from an integrated sustainability perspectiveEditorial Note: Parts of this Peer Review File have been redacted as indicated to remove third-party material where no permission to publish could be obtained.

REVIEWER COMMENTS

Reviewer #1 (Remarks to the Author):

I have added my comments to the attachment as an Word File.

Review report

This manuscript under the title of “Global transboundary synergies and trade-offs¹ among sustainable development goals” tries to check how each Sustainable Development Goal of the United Nations interact with each other. This research can show whether these interactions have a synergistic character or they are trade-off. The former would suggest a global-flow perspective, while the later recommends a local-flow view-point. This manuscript has addressed a critical, interesting, and hot topic, the spillover effects of sustainability. Therefore, I recommend to publish this manuscript after some major revisions, described below.

One the most important limitations of this manuscript is the ignorance of the previous studies working on this topic and also disregarding the use of specific terminologies and perspectives of the sustainability literature like “integrated sustainability”, “weak sustainability”, “strong sustainability”, “sustainable development pillars (or dimensions) including social, environment, and economy, as well as sustainability pillars including 3Ps, People, Planet, and Prosperity. I give you a figure which can help you in this term.

[redacted]

Figure 5. Various sustainability concepts according to the literature review
Source: "Sustainable development goals and transportation modes: Analyzing sustainability pillars of environment, health, and economy

Although the title has appropriately some critical words in this topic like “synergies”, “trade-off”, and “sustainable development goals, it can add the term of “integrated sustainability” as a center for the research. This perspective gives a core to the title, covering both the concepts of “synergies” and “trade-off”.

Regarding the following statement, I propose a comment:

“In contrast, SDG interactions between neighbouring countries were 1.54 times stronger than those between distant countries owing to the transboundary flows in nature (including river flow, ocean currents, and air flow).”

The author can add another important factor to their analysis. Sea and ocean distances are comparative different from land distances in terms of disseminating sustainable development. Accordingly, sea and ocean distances have more capability to transfer merchandise and technology since maritime transportation have more capacity for transferring them, compared with land transportation. A paper has accepted this analysis: “Sustainable development goals: transportation, health and public policy”.

The above approach can be merged into the classification of this research in terms of “human-caused flows” and “nature-caused flows”.

The fourth paragraph of the introduction section can add a topic sentence to propose the core of the paragraph. This topic sentence can use the concept of “integrated sustainability perspective”. For example, this research checks the “integrated sustainability perspective” regarding SDGs to find if each SDG has a spatially and contextually synergistic or trade-off spillover effect on the other SDGs. The integrated sustainability perspective has been described in the following paper: “Sustainable development spillover effects between North America and MENA: Analyzing the integrated sustainability perspective”. Actually, this perspective can join this research into a main current of literature to increase its effectiveness.

In addition, this manuscript should go beyond the spillover effects of SDGs in its literature review. The literature review should show that the concept of spillover effects works not only among SDGs but also between the three pillars of sustainable development including social, environment, and economy. More specifically, dealing with these pillars can improve the problem statement and research question by defining the perspectives of “weak sustainability” and “strong sustainability”. For finding these concepts the authors can various papers like “Environmental pollution, economic growth, population, industrialization, and technology in weak and strong sustainability: using STIRPAT model”.

Figure 1a and 1b should find two harmonized captions. An example is as follows but the authors can find a more appropriate substitute.

- a- Human-caused flows
- b- Nature-effect flows

Just like you’re the resulted figure of this paper in Figure 2 about the spillover effects of SDGs, another paper has these results from a different outlook: the spillover effects of sustainability among the three pillars of sustainable development within different regions of the world. Using

these concepts, results, findings, and perspectives can empower this literature. These papers and findings should find and join each other to boost their claims, tests, questions, concepts, and definitions.

[redacted]

Source: Sustainability spillover effects of social, environment and economy: mapping global sustainable development in a systematic analysis

[redacted]

To increase the interconnection of this manuscript with the literature, the research can use the mentioned concepts, propose the results of the previous studies about spillover effects of sustainability, and compare its findings with the results of other researchers in the discussion or result section.

The abstract has empty rooms for further improvement. It has some limitations and language errors.

This manuscript has serious problems regarding both the colloquial and academic language use as some sentences are unclear. In addition, it has some errors regarding the grammar, spelling, and syntax. The manuscript should be rechecked carefully with language in mind. The following revision is an example:

Original: Here, we quantified the transboundary SDG interactions between 1260 pairs of SDG indicators across 156 countries in 2020.

Revised: This research quantifies the transboundary SDG interactions between 1260 pairs of SDG indicators across 156 countries in 2020.

Original: In this study, we propose a conceptual framework ...

Revised: This study proposes a conceptual framework ...

The manuscript can add two more element to the analysis or discussion: whether the results are promoting the concept of “globalization” or “de-globalization”. Whther the results suggest the “flow-based governance” or “local-based governance”! These concepts have been mentioned in the previous studies and this manuscript can use them to find a stronger interconnection with the literature.

Reviewer #2 (Remarks to the Author):

This paper investigates the correlations of the United Nations Sustainable Development Goals (SDGs) among countries. The main finding is that over two-thirds of the countries' SDGs are positively correlated. Furthermore, the paper states that the interactions via international trade were stronger between distant countries than between neighboring countries, while the interactions via flows in nature (like via rivers, oceans, or air) were stronger for neighboring countries.

While I like the research idea to investigate the relationship between SDGs among countries, my impression is that the analysis is very descriptive in nature. The analysis is based on constructing a spatial weights matrix and calculating Moran's I. Moran's I was invented as a useful statistic to detect spatial correlation based on residuals from a least-squares regression. However, as the test is performed in the current paper, it is completely unconditional, i.e., does not control for other explanatory variables. Furthermore, given today's computation power and software, the estimation of a spatial regression model is relatively simple and would allow for inference on the spatial dependence parameter. See for example LeSage, James P. / Pace, R. Kelly (2009), *Introduction to Spatial Econometrics*, CRC Press, Taylor & Francis Group.

The different channels are distinguished by constructing different weight matrices. In doing so, the weights are taken as exogenous. With that, the results are suggestive, but cannot directly claim to demonstrate the actual channel at work. Many things can for example be correlated with international trade. Therefore, many papers used clean identification strategies and/or theoretical frameworks to quantify the effects. For CO₂ emissions, for example, there is a comparatively large literature that tries to quantify carbon leakage, i.e., how environmental policies in one country change relative goods prices and hence shift production of CO₂-intensive goods to places that are exempt from such regulations and therefore counteract parts of the CO₂ reductions due to the environmental policy (see, for example, Aichele, Rahel / Felbermayr, Gabriel (2015), *Kyoto and carbon leakage: An empirical analysis of the carbon content of bilateral trade*, *Review of Economics and Statistics*, Vol. 97, No. 1, p. 104-115, Baylis, Kathy / Fullerton, Don / Karney, Daniel H. (2013), *Leakage, welfare, and cost-effectiveness of carbon policy*, *American Economic Review*, Vol. 103, No. 3, pp. 332-337.) See also the works by Joseph Shapiro and co-authors (<https://joseph-s-shapiro.com/>) or Esteban Rossi-Hansberg and co-authors (https://rossihansberg.economics.uchicago.edu/research_topic_Environmental.html).

Taking the spatial weights matrix as exogenous and ignoring potential endogeneity concerns may lead to biased estimates. The literature suggested various ways to deal with potential endogenous weights, such as instrumenting the endogenous weights (Kelejian and Piras (2014), *Estimation of spatial models with endogenous weighting matrices, and an application to a demand model for cigarettes*, *Regional Science and Urban Economics*, 2014, vol. 46, issue C, 140-149; Qu, Wang, and Lee (2016); *Instrumental variable estimation of a spatial dynamic panel model with endogenous spatial weights when T is small*, *The Econometrics Journal*, Volume 19, Issue 3, 1 October 2016, Pages 261–290,) or using a control function approach (Qu and Lee (2015); *Estimating a spatial autoregressive model with an endogenous spatial weight matrix*, *Journal of Econometrics*, 2015, vol. 184, issue 2, 209-232; Qu, Lee, and Yu (2017), *QML estimation of spatial dynamic panel data models with endogenous time varying spatial weights matrices*, *Journal of Econometrics*, Volume 197, Issue 2, Pages 173-201; Qu, Lee, and Yang (2021), *Estimation of a SAR model with endogenous spatial weights constructed by bilateral variables*, *Journal of Econometrics*, Volume 221, Issue 1, Pages 180-197).

Additionally, the different channels may be at work at the same time. However, each spatial correlation is tested independently from each other. In a spatial model, various weights matrices could be included at the same time (see Gupta, Abhimanyu / Robinson, Peter M. (2015), *Inference on higher-order spatial autoregressive models with increasingly many parameters*, *Journal of Econometrics*, Vol. 186, No. 1, pp. 19-31, for example), i.e., a higher-order spatial regression model could be specified. In this way, the different channels could be captured at the same time and their relative influences

could be investigated.

Reviewer #3 (Remarks to the Author):

This manuscript analyzes how SDGs are interrelated, and thus contributes to a recently fast growing body of literature. The specific contribution of this article is that it does not focus on the interactions between different SDGs only, but on how they influence each other across countries. This is innovative as it tackles the important issue that countries need to collaborate in order to achieve SDGs, and take extra-territorial spillovers into account. However, there are some important shortcomings and simplifications in this article that question the relevancy of findings. Given that, I have my doubts if the article is innovative enough to warrant publication in Nature Communications. Below are some important critical points that the authors need to address, in any case.

- "Various channels" is used to describe the different types of links. This illustrates quite well one of the problems with this paper: the feeling that these pieces of information were included because they are available. Of course, one has to work with available data, and the most relevant information is sometimes not available, but there is a clear lack of theoretical and conceptual thinking in the paper about what these ties are, and what they are not, and how they fit together.
- Is the distinction between distant and neighboring countries simply dichotomous? That is a clear limitation, as many of the ties represented in the "channels" clearly depend on distance, and not on neighborhood / joint border status only. I would encourage the authors to include a measure of actual distance, somehow (potentially different thresholds would work...), and / or at least discuss this issue very critically.
- Are the synergy links (as well as the different flows) based on a directed or undirected understanding of network ties? Please clarify and discuss critically (regarding the information that might be lost if working with undirected ties).
- I don't think the figures, e.g. figure 4, are very easy to read. Why are low-income countries in an outer circle? What do the circles mean, anyways? I don't think working with circles is very intuitive in this case and I would encourage the authors to rethink this.
- Crucially, SDG interactions between countries can result in mutual SDG achievement, if they are synergistic. However, what if one country does work "against" SDG achievement for some reason? Then, both countries – given the interactions – would mutually fail, right? This is different from non-synergistic interactions (where one country's action towards SDG would result in another country's action against an SDG). So synergies can also result in vicious circles if the "synergy" develops in "negative" ways, that is, against SDG achievement. Meaning that if country A fails, country B also fails...In the article related to the Pham-Truffert et al. database, there's some discussion about this. This is crucial for the interpretation of results in this article as well, so the authors need to discuss this. It can change the interpretation of findings quite a bit.
- The text at the beginning of the results section is not straightforward – it is never clear to what total the percentages relate to. Please clarify.
- The discussion is quite thin. The general statement that "the analysis shows that countries depend on each other" is a natural result given the research design, not a specific finding. I would encourage the authors to be more specific in their discussion, and to improve the link of the discussion elements to the literature.
- Related to this, I would like to see more discussion on how some findings are related, and potentially interpreted. E.g., are there any potential explanations on why high-income countries have stronger interactions with neighboring countries than with distant ones? Also, it's not surprising that natural flows result in stronger relations between neighboring countries than with distant countries (e.g. flows of water and air). Can the authors open up the discussion a bit more and discuss which types of natural flows might behave similarly, or differently, than those included in the empirics?
- Also, in the discussion and elsewhere, there is a strong emphasis on Tobler's first law of geography. While not being a specialist in the field of geography, I would be highly surprised if that law had not been questioned elsewhere, and earlier, in other contexts. So I don't really understand the strong

focus of the authors on their results contradicting this law. I would put less emphasis on that. Plus the references to that law and the literature relying on it (and probably sometimes also contradicting it) are lacking.

- I think the different methodological steps in the methods part would strongly benefit from an example so that the reader can follow the calculations through a specific example. Ideally the same example is used for all subsequent steps.

Point-to-Point Response to Reviewer's comments

Reviewer #1

1. This manuscript under the title of “Global transboundary synergies and trade-offs among sustainable development goals” tries to check how each Sustainable Development Goal of the United Nations interact with each other. This research can show whether these interactions have a synergistic character or they are trade-off. The former would suggest a global-flow perspective, while the later recommends a local-flow view-point. This manuscript has addressed a critical, interesting, and hot topic, the spillover effects of sustainability. Therefore, I recommend to publish this manuscript after some major revisions, described below.

Re: Thanks a lot for your kind and valuable comments. We have followed your instructions below to revise the manuscript, which has substantially improved the quality of the manuscript.

2. One the most important limitations of this manuscript is the ignorance of the previous studies working on this topic and also disregarding the use of specific terminologies and perspectives of the sustainability literature like “integrated sustainability”, “weak sustainability”, “strong sustainability”, “sustainable development pillars (or dimensions) including social, environment, and economy, as well as sustainability pillars including 3Ps, People, Planet, and Prosperity. I give you a figure which can help you in this term.

[redacted]

Figure 5. Various sustainability concepts according to the literature review

Source: “Sustainable development goals and transportation modes: Analyzing sustainability pillars of environment, health, and economy.

Re: Thanks a lot for your valuable suggestions. We have read and cited previous important studies, and also adopted terminologies about this topic in lines 33-42 in Introduction.

“Sustainability, as discussed in the literature, is often approached from two perspectives: weak and strong sustainability². Weak sustainability posits that each of these pillars holds equal weight and that the pillars are interchangeable^{2,3}. Strong sustainability prioritises the environmental pillar^{2,3}. However, recent research has introduced a new perspective on sustainable development: integrated sustainability³⁻⁵. This concept extends beyond traditional weak and strong sustainability perspectives and incorporates the spillover effects generated by the transboundary interactions across regions as a fourth pillar, alongside the original three pillars⁵. These spillover effects represent the interplay of the three original pillars of sustainable development between two or more regions^{4,6}. In the current interconnected world, transboundary

interactions across countries may positively or negatively affect SDGs in various other countries⁷.”

We have discussed sustainability pillars (social, environment, economy, and the spillover effect) and 4Ps (People, Planet, Prosperity, and Peace and partnership) in lines 264-271 in Discussion.

“This assessment was conducted through various channels, including international trade, river flow, ocean currents, and air flow, by employing an integrated sustainability perspective³. This study makes a key contribution by quantifying the magnitude and direction of the fourth pillar of integrated sustainability: the spillover effects caused by human–nature interactions. Therefore, sustainable development, as considered in this study, rests on four pillars: 1) social, 2) environmental, 3) economic, and 4) spillover effects³. These pillars correlate with four key principles: 1) people, 2) planets, 3) prosperity, and 4) peace and partnership^{3,5}.”

References:

2. Nasrollahi, Z., Hashemi, M., Bameri, S. & Mohamad Taghvaei, V. *Environmental pollution, economic growth, population, industrialization, and technology in weak and strong sustainability: using STIRPAT model. Environ Dev Sustain* **22**, 1105–1122 (2020).
3. Taghvaei, V. M., Nodehi, M., Saber, R. M. & Mohebi, M. *Sustainable development goals and transportation modes: Analyzing sustainability pillars of environment, health, and economy. World Development Sustainability* **1**, 100018 (2022).
4. Taghvaei, V. M., Nodehi, M., Arani, A. A., Jafari, Y. & Shirazi, J. K. *Sustainability spillover effects of social, environment and economy: mapping global sustainable development in a systematic analysis. Asia-Pac J Reg Sci* **7**, 329–353 (2023).
5. Mohamad Taghvaei, V., Assari Arani, A. & Agheli, L. *Sustainable development spillover effects between North America and MENA: Analyzing the integrated sustainability perspective. Environmental and Sustainability Indicators* **14**, 100182 (2022).
6. Engström, R. E. et al. *Succeeding at home and abroad: accounting for the international spillovers of cities’ SDG actions. npj Urban Sustain* **1**, 1–5 (2021).
7. Pham-Truffert, M., Metz, F., Fischer, M., Rueff, H. & Messerli, P. *Interactions among Sustainable Development Goals: Knowledge for identifying multipliers and virtuous cycles. Sustainable development* **28**, 1236–1250 (2020).

3. Although the title has appropriately some critical words in this topic like “synergies”, “trade-off”, and “sustainable development goals, it can add the term

of “integrated sustainability” as a center for the research. This perspective gives a core to the title, covering both the concepts of “synergies” and “trade-off”.

Re: Thanks for your insightful idea. We have revised the title by including this key word, integrated sustainability, in the title, as shown below:

“Global Transboundary Synergies and Trade-offs Among Sustainable Development Goals From An Integrated Sustainability Perspective”

4. Regarding the following statement, I propose a comment: “In contrast, SDG interactions between neighbouring countries were 1.54 times stronger than those between distant countries owing to the transboundary flows in nature (including river flow, ocean currents, and air flow).”

The author can add another important factor to their analysis. Sea and ocean distances are comparative different from land distances in terms of disseminating sustainable development. Accordingly, sea and ocean distances have more capability to transfer merchandise and technology since maritime transportation have more capacity for transferring them, compared with land transportation. A paper has accepted this analysis: “Sustainable development goals: transportation, health and public policy”.

Re: That is a great idea and thank you for recommending such a valuable paper! We certainly agree that maritime transport plays an important role in driving transboundary linkages via international shipping networks. Unfortunately, data constraints prevented us from constructing a spatial weight matrix reflecting global maritime connectivity patterns for the present study. While unable to directly operationalize this concept due to limitations, we have highlighted improving quantification of maritime influence as future research in lines 336-348 in Discussion section and cites the valuable paper. We appreciate you pushing our analysis to consider additional pathways of cross-boundary influence beyond what current data allowed.

“In addition to international trade, river flow, ocean currents, and air flow, other cross-border exchanges shape SDG interactions^{49,50}. For example, owing to the high-volume nature of seaborne freight, maritime shipping is well-suited for transporting goods across international borders in regions with extensive coastlines⁵¹. Cargo vessels can accommodate the bulk shipping of diverse goods across long distances in a relatively efficient manner compared with other forms of international transport⁵¹. This strengthens economic cooperation and trade opportunities between coastal countries⁵¹. Future studies should further explore the impacts of various human-caused transboundary flows, such as maritime transportation⁵¹, technological

*transfer, investment, knowledge sharing, human migration, disease dissemination, and information diffusion, once data become available*⁵². *The influence of nature-caused flows, including animal migration, seed dispersal, and disease spread, is also worth investigating*⁵³. *Capturing these additional linkages may provide deeper insights into the complex interconnected relationships between countries' progress towards achieving the SDGs.*”

Reference:

51. Mohamad Taghvaei, V. et al. *Sustainable development goals: transportation, health and public policy. REPS 8, 134–161 (2023).*

5. The above approach can be merged into the classification of this research in terms of “human-caused flows” and “nature-caused flows”.

Re: Thanks for your suggestion. We have included maritime transport as an example in the classification of flows to “human-caused flows” and cited related paper to support it in lines 342-345.

*“Future studies should further explore the impacts of various human-caused transboundary flows, such as **maritime transportation**⁵¹, technological transfer, investment, knowledge sharing, human migration, disease dissemination, and information diffusion, once data become available*⁵¹..... *Capturing these additional linkages may provide deeper insights into the complex interconnected relationships between countries' progress towards achieving the SDGs.”*

Reference:

51. Mohamad Taghvaei, V. et al. *Sustainable development goals: transportation, health and public policy. REPS 8, 134–161 (2023).*

6. The fourth paragraph of the introduction section can add a topic sentence to propose the core of the paragraph. This topic sentence can use the concept of “integrated sustainability perspective”. For example, this research checks the “integrated sustainability perspective” regarding SDGs to find if each SDG has a spatially and contextually synergistic or trade-off spillover effect on the other SDGs. The integrated sustainability perspective has been described in the following paper: “Sustainable development spillover effects between North America and MENA: Analyzing the integrated sustainability perspective”. Actually, this perspective can join this research into a main current of literature to increase its effectiveness.

Re: This is a great suggestion. We also have included this concept in the first sentence of the fourth paragraph in the Introduction Section based on your example, as shown below:

“This study investigates integrated sustainability in the context of the SDGs to determine whether each SDG indicator exerts positive or negative spillover effects on the others and to quantify these effects.”

This valuable paper has been cited in the first paragraph of the Introduction, as shown below:

“Sustainability, as discussed in the literature, is often approached from two perspectives: weak and strong sustainability². Weak sustainability posits that each of these pillars holds equal weight and that the pillars are interchangeable^{2,3}. Strong sustainability prioritises the environmental pillar^{2,3}. However, recent research has introduced a new perspective on sustainable development: integrated sustainability³⁻⁵. This concept extends beyond traditional weak and strong sustainability perspectives and incorporates the spillover effects generated by the transboundary interactions across regions as a fourth pillar, alongside the original three pillars⁵.

References:

5. Mohamad Taghvaei, V., Assari Arani, A. & Agheli, L. Sustainable development spillover effects between North America and MENA: Analyzing the integrated sustainability perspective. *Environmental and Sustainability Indicators* 14, 100182 (2022).

We have also revised the title by including this concept in the title, as shown below:

“Global Transboundary Synergies and Trade-offs Among Sustainable Development Goals From An Integrated Sustainability Perspective”

7. In addition, this manuscript should go beyond the spillover effects of SDGs in its literature review. The literature review should show that the concept of spillover effects works not only among SDGs but also between the three pillars of sustainable development including social, environment, and economy. More specifically, dealing with these pillars can improve the problem statement and research question by defining the perspectives of “weak sustainability” and “strong sustainability”. For finding these concepts the authors can various papers like “Environmental pollution, economic growth, population,

industrialization, and technology in weak and strong sustainability: using STIRPAT model”.

Re: Thanks for your suggestion. We have cited related papers and included the statement that spillover effects not only work among SDGs but also between the three pillars in Introduction. In addition, the perspectives of “weak sustainability” and “strong sustainability” have been explained in Introduction.

*“The three main pillars of sustainability—economy, society, and environment—are encompassed within these goals. Sustainability, as discussed in the literature, is often approached from two perspectives: weak and strong sustainability². **Weak sustainability** posits that each of these pillars holds equal weight and that the pillars are interchangeable^{2,3}. **Strong sustainability** prioritises the environmental pillar^{2,3}. However, recent research has introduced a new perspective on sustainable development: **integrated sustainability**³⁻⁵. This concept extends beyond traditional weak and strong sustainability perspectives and incorporates the spillover effects generated by the transboundary interactions across regions as a fourth pillar, alongside the original three pillars⁵. **These spillover effects represent the interplay of the three original pillars of sustainable development between two or more regions^{4,6}.**”*

References:

2. Nasrollahi, Z., Hashemi, M., Bameri, S. & Mohamad Taghvaei, V. *Environmental pollution, economic growth, population, industrialization, and technology in weak and strong sustainability: using STIRPAT model. Environ Dev Sustain* 22, 1105–1122 (2020).
3. Taghvaei, V. M., Nodehi, M., Saber, R. M. & Mohebi, M. *Sustainable development goals and transportation modes: Analyzing sustainability pillars of environment, health, and economy. World Development Sustainability* 1, 100018 (2022).
4. Taghvaei, V. M., Nodehi, M., Arani, A. A., Jafari, Y. & Shirazi, J. K. *Sustainability spillover effects of social, environment and economy: mapping global sustainable development in a systematic analysis. Asia-Pac J Reg Sci* 7, 329–353 (2023).
5. Mohamad Taghvaei, V., Assari Arani, A. & Agheli, L. *Sustainable development spillover effects between North America and MENA: Analyzing the integrated sustainability perspective. Environmental and Sustainability Indicators* 14, 100182 (2022).
6. Engström, R. E. et al. *Succeeding at home and abroad: accounting for the international spillovers of cities’ SDG actions. npj Urban Sustain* 1, 1–5 (2021).

7. *Pham-Truffert, M., Metz, F., Fischer, M., Rueff, H. & Messerli, P. Interactions among Sustainable Development Goals: Knowledge for identifying multipliers and virtuous cycles. Sustainable development 28, 1236–1250 (2020).*

8. Figure 1a and 1b should find two harmonized captions. An example is as follows but the authors can find a more appropriate substitute. a-Human-caused flows. b-Nature-effect flows.

Re: These two terms indeed look more harmonized. We have revised the captions in Figure 1 to human-caused flows and nature-caused flows.

Figure 1. Conceptual framework of transboundary interactions of sustainable development goals (SDG) across countries.

9. Just like you're the resulted figure of this paper in Figure 2 about the spillover effects of SDGs, another paper has these results from a different outlook: the spillover effects of sustainability among the three pillars of sustainable development within different regions of the world. Using these concepts, results, findings, and perspectives can empower this literature. These

papers and findings should find and join each other to boost their claims, tests, questions, concepts, and definitions.

[redacted]

Source: Sustainability spillover effects of social, environment and economy: mapping global sustainable development in a systematic analysis

[redacted]

To increase the interconnection of this manuscript with the literature, the research can use the mentioned concepts, propose the results of the previous studies about spillover effects of sustainability, and compare its findings with the results of other researchers in the discussion or result section.

Re: Thanks a lot for providing the interesting findings in these papers. We have used the concepts you mentioned in lines 36-39 in Introduction.

“However, recent research has introduced a new perspective on sustainable development: integrated sustainability³⁻⁵. This concept extends beyond traditional weak and strong sustainability perspectives and incorporates the spillover effects

generated by the transboundary interactions across regions as a fourth pillar, alongside the original three pillars⁵.”

References:

3. *Taghvaei, V. M., Nodehi, M., Saber, R. M. & Mohebi, M. Sustainable development goals and transportation modes: Analyzing sustainability pillars of environment, health, and economy. World Development Sustainability 1, 100018 (2022).*
4. *Taghvaei, V. M., Nodehi, M., Arani, A. A., Jafari, Y. & Shirazi, J. K. Sustainability spillover effects of social, environment and economy: mapping global sustainable development in a systematic analysis. Asia-Pac J Reg Sci 7, 329–353 (2023).*
5. *Mohamad Taghvaei, V., Assari Arani, A. & Agheli, L. Sustainable development spillover effects between North America and MENA: Analyzing the integrated sustainability perspective. Environmental and Sustainability Indicators 14, 100182 (2022).*

We have included the results of previous studies, and compared them with our findings in lines 272-283 and lines 293-312 in Discussion section.

“Some research credits globalisation and openness with benefiting sustainability and economic development by invoking Ricardo’s theory of comparative advantage^{4,5,35}. However, other scholars argue that openness and globalisation run contrary to sustainability goals, based on the pollution haven hypothesis^{5,36,37}. While theories of comparative advantage highlight the welfare gains of interconnectedness, the pollution haven hypothesis introduces an important caveat regarding potential cross-border regulatory distortions and their impact on sustainability outcomes in integrated economies at different development levels. The study demonstrate that transboundary synergistic effects through international trade were the dominant form of interaction across borders, revealing the overall positive impact of globalisation and openness on advancing global sustainability in an interconnected world. While trade-offs exist in some issues, the predominance of cross-border coordination benefits underscores how collective progress is enhanced through continued cooperation at a global scale.”

“Tobler’s first law of geography—that everything is related to everything else, but near things are more related than distant ones—serves as a foundational principle in numerous research fields, including spatial analysis in epidemiology³⁸, crime pattern analysis³⁹, economic development^{40,41}, and environmental issues⁴². The continuing relevance and applicability of this law are evidenced by the broad range of methodologies and concepts that have been developed based on it. In modern times,

as sustainability issues increasingly intersect with geographical considerations, there has been a growing interest in revisiting and further exploring Tobler's law, especially with respect to SDGs⁴³. While the law's core principle remains valid in many instances, studies have illuminated the complex ways in which distance can shape interrelations, sometimes counterintuitively¹⁶. For instance, some studies have highlighted habitat losses triggered by distant consumption through international trade^{44,45}. The negative effects of distant activities on sustainable fisheries further challenge this geographical principle^{46,47}. The results of this study suggested that the synergistic effects were 5.65% more pronounced in interactions between trade partners that did not share borders compared with those between neighbouring counterparts. Due to globalisation, different countries have become more connected and less geographically limited through international trade. Globally, non-neighbouring countries can benefit from comparative advantages by diversifying their traded goods and services, allowing them to interact more with each other than with neighbouring countries. While Tobler's law remains valuable for understanding geographical influences, recent research has revealed the importance of considering a broader array of factors, including non-proximate influences.”

References:

4. Taghvaei, V. M., Nodehi, M., Arani, A. A., Jafari, Y. & Shirazi, J. K. Sustainability spillover effects of social, environment and economy: mapping global sustainable development in a systematic analysis. *Asia-Pac J Reg Sci* 7, 329–353 (2023).
5. Mohamad Taghvaei, V., Assari Arani, A. & Agheli, L. Sustainable development spillover effects between North America and MENA: Analyzing the integrated sustainability perspective. *Environmental and Sustainability Indicators* 14, 100182 (2022).
16. Xu, Z. et al. Impacts of international trade on global sustainable development. *Nature Sustainability* 3, 964–971 (2020).
35. Umar, M., Ji, X., Kirikkaleli, D., Shahbaz, M. & Zhou, X. Environmental cost of natural resources utilization and economic growth: can China shift some burden through globalization for sustainable development? *Sustainable Development* 28, 1678–1688 (2020).
36. Cole, M. A. & Elliott, R. J. R. Determining the trade–environment composition effect: the role of capital, labor and environmental regulations. *Journal of Environmental Economics and Management* 46, 363–383 (2003).
37. Cave, L. A. & Blomquist, G. C. Environmental policy in the European Union: Fostering the development of pollution havens? *Ecological Economics* 65, 253–261 (2008).
38. Ulimwengu, J. & Kibonge, A. Spatial spillover and COVID-19 spread in the U.S. *BMC Public Health* 21, 1765 (2021).

39. Carboni, O. A. & Detotto, C. *The economic consequences of crime in Italy. Journal of Economic Studies* **43**, 122–140 (2016).
40. Peng, W., Yin, Y., Kuang, C., Wen, Z. & Kuang, J. *Spatial spillover effect of green innovation on economic development quality in China: Evidence from a panel data of 270 prefecture-level and above cities. Sustainable Cities and Society* **69**, 102863 (2021).
41. Marinos, T., Belegri-Roboli, A., Michaelides, P. G. & Konstantakis, K. N. *The spatial spillover effect of transport infrastructures in the Greek economy (2000–2013): A panel data analysis. Research in Transportation Economics* **94**, 101179 (2022).
42. Nan, S., Huo, Y., You, W. & Guo, Y. *Globalization spatial spillover effects and carbon emissions: What is the role of economic complexity? Energy Economics* **112**, 106184 (2022).
43. Manning, N., Li, Y. & Liu, J. *Broader applicability of the metacoupling framework than Tobler’s first law of geography for global sustainability: a systematic review. Geography and Sustainability* (2022).
44. Green, J. M. et al. *Linking global drivers of agricultural trade to on-the-ground impacts on biodiversity. Proceedings of the National Academy of Sciences* **116**, 23202–23208 (2019).
45. Lenzen, M. et al. *International trade drives biodiversity threats in developing nations. Nature* **486**, 109–112 (2012).
46. Carlson, A. K., Taylor, W. W., Rubenstein, D. I., Levin, S. A. & Liu, J. *Global marine fishing across space and time. Sustainability* **12**, 4714 (2020).
47. Carlson, A. K., Taylor, W. W. & Liu, J. *Using the telecoupling framework to improve Great Lakes fisheries sustainability. Aquatic ecosystem health & management* **22**, 342–354 (2019).

10. The abstract has empty rooms for further improvement. It has some limitations and language errors.

Re: Thank you for your valuable feedback. We've taken your comments into account and have thoroughly revised and improved the abstract. We have also engaged native English speakers at Editage to proofread the the whole manuscript, as evidenced by the attached certification.

The updated abstract, incorporating all these changes, is as follows:

“Domestic attempts to advance the United Nations Sustainable Development Goals (SDGs) in a country can have synergistic and/or trade-off effects on the advancement of SDGs in other countries. Transboundary SDG interactions between countries can be delivered through various transmission channels (e.g., international trade, river flow, ocean currents, and air flow). However, studies on how the SDG advancements of one country interact with those of other countries through these channels are

scarce. This study quantified the transboundary SDG interactions between 768 pairs of SDG indicators across 121 countries from 2010 to 2020. A spatial interaction index was developed to represent the overall magnitude of the transboundary SDG interactions. The results showed that transboundary synergistic effects were 3.82 times stronger than trade-off effects, indicating that transboundary SDG interactions between the countries can promote SDG achievement. Although high income countries only comprised 14.18% of the global population, they contributed considerably to total SDG interactions worldwide (61.22% of the accumulated interactions), indicating a greater responsibility for SDG achievement in other countries. Transboundary synergistic effects via international trade were 5.65% more pronounced in interactions with trade partners outside their immediate geographic vicinity compared to interactions with neighbouring trade partners. Conversely, through nature-caused flow channels (including river flow, ocean currents, and air flow), neighbouring countries experienced transboundary synergistic effects that were 36.75% stronger than those observed between non-neighbouring countries. These findings can be used to capitalise on transboundary SDG synergies and minimise trade-offs between countries to achieve SDGs worldwide.”

11. This manuscript has serious problems regarding both the colloquial and academic language use as some sentences are unclear. In addition, it has some errors regarding the grammar, spelling, and syntax. The manuscript should be rechecked carefully with language in mind. The following revision is an example:

Original: Here, we quantified the transboundary SDG interactions between 1260 pairs of SDG indicators across 156 countries in 2020.

Revised: This research quantifies the transboundary SDG interactions between 1260 pairs of SDG indicators across 156 countries in 2020.

Original: In this study, we propose a conceptual framework ...

Revised: This study proposes a conceptual framework ...

Re: Many thanks for this suggestion. We have followed your suggestion to revise these sentences and made appropriate revisions to improve objectivity throughout the manuscript. Specifically, we have reviewed instances where "we" was used to focus on the research aims and findings rather than actors. In addition, we have engaged native English speakers at Editage to proofread the the whole manuscript, as evidenced by the attached certification.

12. The manuscript can add two more element to the analysis or discussion: whether the results are promoting the concept of “globalization” or “de-globalization”. Whether the results suggest the “flow-based governance” or “local-based governance”! These concepts have been mentioned in the previous studies and this manuscript can use them to find a stronger interconnection with the literature.

Re: Thanks for your suggestions. We have cited related studies and added the elements, globalization, de-globalization, flow-based governance, and local-based governance, in lines 272-283 and lines 313-320 in the discussion.

“Some research credits globalisation and openness with benefiting sustainability and economic development by invoking Ricardo’s theory of comparative advantage^{4,5,35}. However, other scholars argue that openness and globalisation run contrary to sustainability goals, based on the pollution haven hypothesis^{5,36,37}. While theories of comparative advantage highlight the welfare gains of interconnectedness, the pollution haven hypothesis introduces an important caveat regarding potential cross-border regulatory distortions and their impact on sustainability outcomes in integrated economies at different development levels. The study demonstrate that

transboundary synergistic effects through international trade were the dominant form of interaction across borders, revealing the overall positive impact of globalisation and openness on advancing global sustainability in an interconnected world. While trade-offs exist in some issues, the predominance of cross-border coordination benefits underscores how collective progress is enhanced through continued cooperation at a global scale.”

“Transboundary SDG interactions are global issues that transcend individual nations. Beyond traditional place-based governance approaches with a focus on a country’s territory, it is significant to adopt a flow-based perspective. This considers each country in the context of its associations with others by identifying, monitoring, and managing areas where key flows originate, progress between borders, and ultimately terminate^{4,5,48}. This study advocates that countries collaborate to find solutions through international organisations that act as bridges to facilitate global policymaking and support the achievement of the 2030 Agenda. Some international organisations (e.g., the UN and World Trade Organization) were formed to implement adequate measures to address global issues.”

References:

4. Taghvaei, V. M., Nodehi, M., Arani, A. A., Jafari, Y. & Shirazi, J. K. Sustainability spillover effects of social, environment and economy: mapping global sustainable development in a systematic analysis. *Asia-Pac J Reg Sci* 7, 329–353 (2023).
5. Mohamad Taghvaei, V., Assari Arani, A. & Agheli, L. Sustainable development spillover effects between North America and MENA: Analyzing the integrated sustainability perspective. *Environmental and Sustainability Indicators* 14, 100182 (2022).
35. Umar, M., Ji, X., Kirikkaleli, D., Shahbaz, M. & Zhou, X. Environmental cost of natural resources utilization and economic growth: can China shift some burden through globalization for sustainable development? *Sustainable Development* 28, 1678–1688 (2020).
36. Cole, M. A. & Elliott, R. J. R. Determining the trade–environment composition effect: the role of capital, labor and environmental regulations. *Journal of Environmental Economics and Management* 46, 363–383 (2003).
37. Cave, L. A. & Blomquist, G. C. Environmental policy in the European Union: Fostering the development of pollution havens? *Ecological Economics* 65, 253–261 (2008).
48. Liu, J. et al. Spillover systems in a telecoupled Anthropocene: typology, methods, and governance for global sustainability. *Current Opinion in Environmental Sustainability* 33, 58–69 (2018).

Reviewer #2

1. This paper investigates the correlations of the United Nations Sustainable Development Goals (SDGs) among countries. The main finding is that over two-thirds of the countries' SDGs are positively correlated. Furthermore, the paper states that the interactions via international trade were stronger between distant countries than between neighboring countries, while the interactions via flows in nature (like via rivers, oceans, or air) were stronger for neighboring countries.

Re: We greatly appreciate your insightful comments which substantially enhance strength of methods and quantitative analysis! We have carefully revised the manuscript based on your helpful guidance. Please find our changes detailed below.

1. While I like the research idea to investigate the relationship between SDGs among countries, my impression is that the analysis is very descriptive in nature. The analysis is based on constructing a spatial weights matrix and calculating Moran's I. Moran's I was invented as a useful statistic to detect spatial correlation based on residuals from a least-squares regression. However, as the test is performed in the current paper, it is completely unconditional, i.e., does not control for other explanatory variables. Furthermore, given today's computation power and software, the estimation of a spatial regression model is relatively simple and would allow for inference on the spatial dependence parameter. See for example LeSage, James P. / Pace, R. Kelly (2009), *Introduction to Spatial Econometrics*, CRC Press, Taylor & Francis Group.

Re: Thank you very much for the constructive comments! We have read and cited this very useful recommended book. To enhance the quantitative rigor and methods, we have updated our modeling approach as you advised. Specifically, we now employ the spatial econometric framework proposed by Qu, Wang, and Lee (2016) to estimate transboundary interactions.

Within this spatial regression context, we control for other explanatory factors relating to each SDG indicator, along with the rationale for their selection, in Table A.6, titled 'Explanatory variables and selection rationale for 55 SDG indicators.' Given that this table spans 37 pages, we've refrained from including it here. For a comprehensive view of Table A. 6, we kindly encourage you to refer to the supplemental information provided.

We have changed the quantification method to spatial econometric model in lines 454-492 in Method section:

“Spatial econometric models were used to explore the transboundary SDG

interactions of SDG indicator pairs under different transmission channels⁶².....Based on Qu and Lee (2015)⁷⁰, this study used a control function approach to handle the possible endogenous spatial weight matrix related to international trade flows. The first stage of the 2SIV was estimated using the following regression model⁶⁸:

$$\ln F_{it} = \eta_i + \rho_1 \ln G_{i,t-1} + X_{1it} \gamma + \varepsilon_{it}, \quad t=1,2,\dots,T$$

F_{it} denotes the trade flow in country i in the t th year. γ is a vector of the coefficients of explanatory variables. X_{1it} is a list variables measuring the economy, population, government effectiveness, access to the internet, performance of export sectors, and technological level of country i at time t ⁷³. The indicators used to represent these variables and their data sources are presented in Table A. 5. ρ_1 is the scalar coefficient. Based on the residual $\hat{\varepsilon}_{it}$ from the first-step estimation, this study considered the following model for the second-stage estimation of the 2SIV⁶⁸:

$$\ln SDG_{imt} = c_i + \rho_2 \ln SDG_{i,t-1} + \lambda_1 \sum_{j \neq i} w_{ijt}^1 \ln SDG_{jnt} + X_{2it} \beta + \delta \hat{\varepsilon}_{it} + v_{it}$$

SDG_{imt} and SDG_{jnt} are a pair of SDG indicators that were determined in Step 1, which respectively indicate the interaction receiver m and generator n . ρ_2 is the scale coefficient. λ_1 is the spatial coefficient which can be used to measure the transboundary SDG interactions under the transmission channel of international trade. w_{ijt}^1 is spatial a weight matrix related to international trade between country i and country j in year t . β is a vector of coefficient of the explanatory variables. X_{2it} denotes the explanatory variables, which are shown in **Table A. 6**. The positive variables were transformed by taking their natural logarithms in the spatial dynamic panel data model.”

References:

62. LeSage, J. & Pace, R. K. *Introduction to Spatial Econometrics*. (Chapman and Hall/CRC, 2009). doi:10.1201/9781420064254.

70. Qu, X. & Lee, L. *Estimating a spatial autoregressive model with an endogenous spatial weight matrix. Journal of Econometrics* 184, 209–232 (2015).

2. The different channels are distinguished by constructing different weight matrices. In doing so, the weights are taken as exogenous. With that, the results are suggestive, but cannot directly claim to demonstrate the actual channel at work. Many things can for example be correlated with international trade. Therefore, many papers used clean identification strategies and/or theoretical frameworks to quantify the effects. For CO₂ emissions, for example, there is a comparatively large literature that tries to quantify carbon leakage, i.e., how environmental policies in one country change relative goods prices and hence shift production of CO₂-intensive goods to places that are exempt from such regulations and therefore countervail parts of the CO₂ reductions due to the environmental policy (see, for example, Aichele, Rahel / Felbermayr, Gabriel (2015), *Kyoto and carbon leakage: An empirical analysis of the carbon content of bilateral trade, Review of Economics and Statistics*, Vol. 97, No. 1, p. 104-115, Baylis, Kathy / Fullerton, Don / Karney, Daniel H. (2013), *Leakage, welfare, and cost-effectiveness of carbon policy, American Economic Review*, Vol. 103, No. 3, pp. 332-337.) See also the works by Joseph Shapiro and co-authors (<https://joseph-s-shapiro.com/>) or Esteban Rossi-Hansberg and co-authors (https://rossihansberg.economics.uchicago.edu/research_topic_Environmental.html).

Re: We appreciate your excellent suggestion to address the endogeneity issues related to treating the spatial weight matrix for international trade as exogenous. We have revised our approach based on the clean identification strategy by adopting a two-stage least-squares (2SLS) estimation of the spatial econometric model as proposed by Qu, Wang, and Lee (2016). This change allows us to better account for potential endogeneity caused by the matrix and further strengthen our analysis.

We have also discussed the solutions to endogeneity issues by including all these useful papers in lines 454-464 in Methods section.

“Endogeneity issues may arise in statistical analyses when an explanatory variable is correlated with an error term, leading to biased and inconsistent estimates. These complexities require specific techniques to ensure accurate and reliable results⁶³⁻⁶⁷. Different techniques can be used to address the endogeneity problem⁶³⁻⁶⁷. Among them, clean identification strategies are often employed to isolate the causal effects of a specific variable or intervention on the outcome of interest.”

References:

63. Baylis, K., Fullerton, D. & Karney, D. H. Leakage, Welfare, and Cost-Effectiveness of Carbon Policy. *American Economic Review* **103**, 332–337 (2013).
64. Aichele, R. & Felbermayr, G. Kyoto and carbon leakage: An empirical analysis of the carbon content of bilateral trade. *Review of Economics and Statistics* **97**, 104–115 (2015).
65. Shapiro, J. S. The Environmental Bias of Trade Policy*. *The Quarterly Journal of Economics* **136**, 831–886 (2021).
66. Shapiro, J. S. & Walker, R. Why Is Pollution from US Manufacturing Declining? The Roles of Environmental Regulation, Productivity, and Trade. *American Economic Review* **108**, 3814–3854 (2018).
67. Antràs, P., Redding, S. J. & Rossi-Hansberg, E. Globalization and Pandemics. *American Economic Review* **113**, 939–981 (2023).

We have changed to 2SLS of spatial econometric model and accompanying explanation in lines 461–482 of the Methods Section.

“To deal with the endogeneity issues caused by the elements of the spatial weight matrix involving socioeconomic indicators^{68–72}, this study applied the two-stage instrumental variable (2SIV) estimation of the spatial econometric model to explore the spatial spillover effects of SDG indicators in a panel dataset⁶⁸. Based on Qu and Lee (2015)⁷⁰, this study used a control function approach to handle the possible endogenous spatial weight matrix related to international trade flows. The first stage of the 2SIV was estimated using the following regression model⁶⁸:

$$\ln F_{it} = \eta_i + \rho_1 \ln G_{i,t-1} + X_{it} \gamma + \varepsilon_{it}, \quad t = 1, 2, \dots, T$$

F_{it} denotes the trade flow in country i in the t th year. γ is a vector of the coefficients of explanatory variables. X_{it} is a list variables measuring the economy, population, government effectiveness, access to the internet, performance of export sectors, and technological level of country i at time t ⁷³. The indicators used to represent these variables and their data sources are presented in Table A. 5. ρ_1 is the scalar coefficient. Based on the residual $\hat{\varepsilon}_{it}$ from the first-step estimation, this study considered the following model for the second-stage estimation of the 2SIV⁶⁸:

$$\ln SDG_{imt} = c_i + \rho_2 \ln SDG_{i,t-1} + \lambda_1 \sum_{j \neq i} w_{ijt}^1 \ln SDG_{jnt} + X_{2it} \beta + \delta \hat{\varepsilon}_{it} + v_{it}$$

SDG_{imt} and SDG_{jnt} are a pair of SDG indicators that were determined in Step 1, which respectively indicate the interaction receiver m and generator n . ρ_2 is the scale coefficient. λ_1 is the spatial coefficient which can be used to measure the transboundary SDG interactions under the transmission channel of international trade. w_{ijt}^1 is spatial a weight matrix related to international trade between country i and country j in year t . β is a vector of coefficient of the explanatory variables. X_{2it} denotes the explanatory variables, which are shown in Table A. 6. The positive variables were transformed by taking their natural logarithms in the spatial dynamic panel data model.”

References:

68. Qu, X., Wang, X. & Lee, L. Instrumental variable estimation of a spatial dynamic panel model with endogenous spatial weights when T is small. *The Econometrics Journal* **19**, 261–290 (2016).
69. Kelejian, H. H. & Piras, G. Estimation of spatial models with endogenous weighting matrices, and an application to a demand model for cigarettes. *Regional Science and Urban Economics* **46**, 140–149 (2014).
70. Qu, X. & Lee, L. Estimating a spatial autoregressive model with an endogenous spatial weight matrix. *Journal of Econometrics* **184**, 209–232 (2015).
71. Qu, X., Lee, L. & Yu, J. QML estimation of spatial dynamic panel data models with endogenous time varying spatial weights matrices. *Journal of Econometrics* **197**, 173–201 (2017).
72. Qu, X., Lee, L. & Yang, C. Estimation of a SAR model with endogenous spatial weights constructed by bilateral variables. *Journal of Econometrics* **221**, 180–197 (2021).
73. World Trade Organization. *World Trade Report 2013: Factors Shaping the Future of World Trade*. (WTO Publications, 2014).

3. Taking the spatial weights matrix as exogenous and ignoring potential endogeneity concerns may lead to biased estimates. The literature suggested various way to deal with potential endogenous weights, such as instrumenting the endogenous weights (Kelejian and Piras (2014), Estimation of spatial models with endogenous weighting matrices, and an application to a demand model for

cigarettes, *Regional Science and Urban Economics*, 2014, vol. 46, issue C, 140-149; Qu, Wang, and Lee (2016); Instrumental variable estimation of a spatial dynamic panel model with endogenous spatial weights when T is small, *The Econometrics Journal*, Volume 19, Issue 3, 1 October 2016, Pages 261–290,) or using a control function approach (Qu and Lee (2015); Estimating a spatial autoregressive model with an endogenous spatial weight matrix, *Journal of Econometrics*, 2015, vol. 184, issue 2, 209-232; Qu, Lee, and Yu (2017), QML estimation of spatial dynamic panel data models with endogenous time varying spatial weights matrices, *Journal of Econometrics*, Volume 197, Issue 2, Pages 173-201; Qu, Lee, and Yang (2021), Estimation of a SAR model with endogenous spatial weights constructed by bilateral variables, *Journal of Econometrics*, Volume 221, Issue 1, Pages 180-197).

Re: Thanks a lot for these valuable papers and useful suggestions. To deal with the endogeneity problems when using spatial matrix related to international trade, we have followed your suggestion to use the econometric model proposed by Qu, Wang, and Lee (2016) and cited all these valuable papers properly, as shown in lines 454-482 in Methods Section. Thank you again for suggesting us to incorporate leading work in this area.

“Spatial econometric models were used to explore the transboundary SDG interactions of SDG indicator pairs under different transmission channels⁶². Endogeneity issues may arise in statistical analyses when an explanatory variable is correlated with an error term, leading to biased and inconsistent estimates. These complexities require specific techniques to ensure accurate and reliable results^{63–67}. Different techniques can be used to address the endogeneity problem^{63–67}. Among them, clean identification strategies are often employed to isolate the causal effects of a specific variable or intervention on the outcome of interest. To deal with the endogeneity issues caused by the elements of the spatial weight matrix involving socioeconomic indicators^{68–72}, this study applied the two-stage instrumental variable (2SIV) estimation of the spatial econometric model to explore the spatial spillover effects of SDG indicators in a panel dataset⁶⁸. Based on Qu and Lee (2015)⁷⁰, this study used a control function approach to handle the possible endogenous spatial weight matrix related to international trade flows. The first stage of the 2SIV was estimated using the following regression model⁶⁸:

$$\ln F_{it} = \eta_i + \rho_1 \ln G_{i,t-1} + X_{it}\gamma + \varepsilon_{it}, \quad t=1,2,\dots,T$$

F_{it} denotes the trade flow in country i in the t th year. γ is a vector of the coefficients of explanatory variables. X_{it} is a list variables measuring the economy, population, government effectiveness, access to the internet, performance of export

sectors, and technological level of country i at time t ⁷³. The indicators used to represent these variables and their data sources are presented in Table A. 5. ρ_1 is the scalar coefficient. Based on the residual $\hat{\varepsilon}_{it}$ from the first-step estimation, this study considered the following model for the second-stage estimation of the 2SIV⁶⁸:

$$\ln SDG_{imt} = c_i + \rho_2 \ln SDG_{i,t-1} + \lambda_1 \sum_{j \neq i} w_{ijt}^1 \ln SDG_{jnt} + X_{2it} \beta + \delta \hat{\varepsilon}_{it} + v_{it}$$

SDG_{imt} and SDG_{jnt} are a pair of SDG indicators that were determined in Step 1, which respectively indicate the interaction receiver m and generator n . ρ_2 is the scale coefficient. λ_1 is the spatial coefficient which can be used to measure the transboundary SDG interactions under the transmission channel of international trade. w_{ijt}^1 is spatial a weight matrix related to international trade between country i and country j in year t . β is a vector of coefficient of the explanatory variables. X_{2it} denotes the explanatory variables, which are shown in Table A. 6. The positive variables were transformed by taking their natural logarithms in the spatial dynamic panel data model.”

References:

68. Qu, X., Wang, X. & Lee, L. Instrumental variable estimation of a spatial dynamic panel model with endogenous spatial weights when T is small. *The Econometrics Journal* **19**, 261–290 (2016).
69. Kelejian, H. H. & Piras, G. Estimation of spatial models with endogenous weighting matrices, and an application to a demand model for cigarettes. *Regional Science and Urban Economics* **46**, 140–149 (2014).
70. Qu, X. & Lee, L. Estimating a spatial autoregressive model with an endogenous spatial weight matrix. *Journal of Econometrics* **184**, 209–232 (2015).
71. Qu, X., Lee, L. & Yu, J. QML estimation of spatial dynamic panel data models with endogenous time varying spatial weights matrices. *Journal of Econometrics* **197**, 173–201 (2017).
72. Qu, X., Lee, L. & Yang, C. Estimation of a SAR model with endogenous spatial weights constructed by bilateral variables. *Journal of Econometrics* **221**, 180–197 (2021).

4. Additionally, the different channels may be at work at the same time. However, each spatial correlation is tested independently from each other. In a spatial model, various weights matrices could be included at the same time (see Gupta, Abhimanyu / Robinson, Peter M. (2015), Inference on higher-order spatial autoregressive models with increasingly many parameters, *Journal of Econometrics*, Vol. 186, No. 1, pp. 19-31, for example), i.e., a higher-order spatial regression model could be specified. In this way, the different channels could be captured at the same time and their relative influences could be investigated.

Re: We completely agree with you that different channels can work simultaneously. Therefore, we have incorporated appropriate citations of these recommended papers to contextualize our modeling enhancements. Next, following your guidance, we have constructed a higher-order spatial econometric model by including various spatial weight matrices in lines 483-492 in Methods section.

“Multiple transmission channels could operate simultaneously^{74–77}. Indicators related to SDG 6 (clean water and sanitation), 14 (life below water), and SDG 11.6 (fine particulate matter) could be affected both through the impacts embodied in international trade and nature-caused flows (transboundary river flow, ocean currents, and air flow). This study employed higher-order spatial econometric models to account for real-world complexity. These models can incorporate more than one spatial weight matrix and, thus, characterise various types of spatial dependence.

Spatial weight matrix w_{ij}^2 was specifically utilised to represent channels related to nature-caused flows. λ_2 is a spatial coefficient used to evaluate transboundary SDG interactions under the transmission channel of nature-caused flows.

$$\ln SDG_{imt} = c_i + \rho_2 \ln SDG_{i,t-1} + \lambda_1 \sum_{j \neq i} w_{ijt}^1 \ln SDG_{jmt} + \lambda_2 \sum_{j \neq i} w_{ij}^2 \ln SDG_{jmt} + X_{2it} \beta + \delta \hat{\epsilon}_{it} + v_{it} ”$$

References:

74. Atella, V., Belotti, F., Depalo, D. & Mortari, A. P. Measuring spatial effects in the presence of institutional constraints: The case of Italian Local Health Authority expenditure. *Regional Science and Urban Economics* **49**, 232–241 (2014).
75. Lacombe, D. J. Does econometric methodology matter? An analysis of public policy using spatial econometric techniques. *Geographical analysis* **36**, 105–118 (2004).
76. Gupta, A. & Robinson, P. M. Inference on higher-order spatial autoregressive models with increasingly many parameters. *Journal of Econometrics* **186**, 19–31 (2015).

77. Gupta, A. & Robinson, P. M. *Pseudo maximum likelihood estimation of spatial autoregressive models with increasing dimension. Journal of Econometrics* **202**, 92–107 (2018).

Reviewer #3

1. This manuscript analyzes how SDGs are interrelated, and thus contributes to a recently fast growing body of literature. The specific contribution of this article is that it does not focus on the interactions between different SDGs only, but on how they influence each other across countries. This is innovative as it tackles the important issue that countries need to collaborate in order to achieve SDGs, and take extra-territorial spillovers into account. However, there are some important shortcomings and simplifications in this article that question the relevancy of findings. Given that, I have my doubts if the article is innovative enough to warrant publication in Nature Communications. Below are some important critical points that the authors need to address, in any case.

Re: Thanks for valuable and constructive comments which substantially improve the manuscript. We have followed your instructions to overcome these shortcomings and enrich the content of the manuscript, as shown in the below detailed revisions.

2. “Various channels” is used to describe the different types of links. This illustrates quite well one of the problems with this paper: the feeling that these pieces of information were included because they are available. Of course, one has to work with available data, and the most relevant information is sometimes not available, but there is a clear lack of theoretical and conceptual thinking in the paper about what these ties are, and what they are not, and how they fit together.

Re: We thank you for this suggestion. You raise a fair point that greater theoretical and conceptual clarity is needed around the various connectivity channels included in our analysis. To address this, we've adopted the metacoupling framework as conceptual guidance and established a set of criteria to guide the selection of suitable transmission channels for SDG indicators. You can find the detailed criteria outlined in lines 428-447 of the Methods section.

“This study is grounded in the metacoupling framework, an integrated conceptual construct examining the human–nature interplay within a coupled human–nature system adjacent to that system and from distant locations^{30,37,43}. This framework encompasses all flow types relevant to human and natural systems. Determination of the spatial weight matrix is guided by four key criteria: (1) Relevance: The chosen transmission channel should mirror the real-world transmission mechanics of the SDG indicator. For example, water-related SDG indicators may be interlinked through transboundary rivers. Consequently, the spatial weight matrix, represented by river flows across countries, was utilised to examine the transboundary interactions of indicators related to SDG 6 (clean water and sanitation). (2) Timeliness:

Transmission channels influenced by socio-economic conditions, such as international trade, are dynamic and frequently change over time. Consequently, data series must be updated regularly, published promptly, and be made available for the most recent years to accurately reflect these changes. (3) Coverage: The data must adequately define the relationships between any two countries included in the study. They should provide a comprehensive understanding of the interactions and connections between these countries, offering a broad scope that does not neglect critical relationships. (4) Data availability and quality: The transmission channel data must represent the most accurate measure of a specific issue. They should be derived from reliable national or international sources, such as national statistical offices or intergovernmental organisations to ensure credibility and reliability. Considering these selection criteria, this study incorporated different flow types that exist across countries: trade flows (human-caused flows), river flows, ocean currents, and air flows (nature-caused flows)."

For some transmission channels that haven't been considered in this study due to data constraints, we have discussed this part as future work in lines 336-348 in Discussion Section.

"In addition to international trade, river flow, ocean currents, and air flow, other cross-border exchanges shape SDG interactions^{26,27}. For example, owing to the high-volume nature of seaborne freight, maritime shipping is well-suited for transporting goods across international borders in regions with extensive coastlines²⁸. Cargo vessels can accommodate the bulk shipping of diverse goods across long distances in a relatively efficient manner compared with other forms of international transport²⁸. This strengthens economic cooperation and trade opportunities between coastal countries²⁸. Future studies should further explore the impacts of various human-caused transboundary flows, such as maritime transportation²⁸, technological transfer, investment, knowledge sharing, human migration, disease dissemination, and information diffusion, once data become available²⁹. The influence of nature-caused flows, including animal migration, seed dispersal, and disease spread, is also worth investigating³⁰. Capturing these additional linkages may provide deeper insights into the complex interconnected relationships between countries' progress towards achieving the SDGs."

3. Is the distinction between distant and neighboring countries simply dichotomous? That is a clear limitation, as many of the ties represented in the "channels" clearly depend on distance, and not on neighborhood / joint border status only. I would encourage the authors to include a measure of actual distance, somehow (potentially different thresholds would work...), and / or at least discuss this issue very critically.

Re: This is a great idea. We have critically discussed this issue under these different situations in Methods section. We agree that the implementation of a distance threshold could serve as a method to classify countries into categories other than neighbouring and distant. This approach, undoubtedly, could yield more profound insights, especially if the areas under study have similar land areas. In this particular study, however, the land areas of the 121 countries examined differ significantly. According to World Bank data, their sizes range from a mere 430 sq. km (Barbados) to a vast 16,376,870 sq. km (Russian Federation). This implies that under a specific threshold, some countries may have none or very few neighbours, particularly if the country's land area is small. In contrast, some countries might have a large number of neighbouring countries. In response to the challenge of defining 'distant countries,' we have revised our terminology throughout the manuscript. We now refer to these countries as 'neighbouring' and 'non-neighbouring' countries, which provides a more precise description, as shown in the below research framework.

Figure 1. Conceptual framework of transboundary interactions of sustainable development goals (SDG) across countries.

We acknowledge this approach as a possible limitation of our study. Your suggestion is indeed compelling. In future research, it would be an excellent idea to compare the effects of transboundary interactions of neighbouring and distant countries by incorporating different distance thresholds. We have critically discussed the classification issue about neighbouring and distant regions as future work in lines 565-572 in Methods Section:

“It is essential to acknowledge that the definitions of neighbouring and distant regions are subject to contextual variations, and there is no universally recognised measure that unequivocally designates a region as “neighbouring” or “distant”. In certain research endeavours that seek to investigate the impacts of channels closely related to distance, distant regions can be identified through the application of various distance thresholds, thereby facilitating a more comprehensive understanding of the subject matter. To gain further insight, future studies should compare the impacts of transboundary interactions using different distance thresholds to distinguish between neighbouring and distant regions.”

4. Are the synergy links (as well as the different flows) based on a directed or undirected understanding of network ties? Please clarify and discuss critically (regarding the information that might be lost if working with undirected ties).

Re: Thanks a lot for your question. These synergy/trade-off links are based on the directed understanding of network ties. This is primarily because the interactions among different SDGs are extracted from an interactive SDG repository which documents causal (directed) relationships across SDG targets covering all 17 SDGs¹. Additionally, we have updated the method used to investigate transboundary interactions between countries. We've transitioned from using Global Moran's I to a spatial econometric model. While Global Moran's I allows us to explore only correlational relationships, our current method, the spatial econometric model, accommodates the directed relationship. It does so by treating interaction receivers as dependent variables and interaction generators as independent variables. We have elaborated on the directed relationship in lines 70-75 in the Introduction section and in lines 454-482 in Method section, as follows:

“This study examined 768 pairs of SDG indicators to evaluate how an individual SDG indicator of a country interacts with other countries’ indicators through different channels³¹⁻³³. The pairs of indicators were identified as having causal relationships (e.g., energy intensity and CO₂ emission intensity indicators as the interaction generator and receiver, respectively), which can be derived from the “interactive repository of SDG interactions in CDEdatablog” database⁷ (Tables A. 1 and A. 2).”

“Spatial econometric models were used to explore the transboundary SDG

interactions of SDG indicator pairs under different transmission channels⁶². Endogeneity issues may arise in statistical analyses when an explanatory variable is correlated with an error term, leading to biased and inconsistent estimates. These complexities require specific techniques to ensure accurate and reliable results⁶³⁻⁶⁷. Different techniques can be used to address the endogeneity problem⁶³⁻⁶⁷. Among them, clean identification strategies are often employed to isolate the causal effects of a specific variable or intervention on the outcome of interest. To deal with the endogeneity issues caused by the elements of the spatial weight matrix involving socioeconomic indicators⁶⁸⁻⁷², this study applied the two-stage instrumental variable (2SIV) estimation of the spatial econometric model to explore the spatial spillover effects of SDG indicators in a panel dataset⁶⁸. Based on Qu and Lee (2015)⁷⁰, this study used a control function approach to handle the possible endogenous spatial weight matrix related to international trade flows. The first stage of the 2SIV was estimated using the following regression model⁶⁸:

$$\ln F_{it} = \eta_i + \rho_1 \ln G_{i,t-1} + X_{1it} \gamma + \varepsilon_{it}, \quad t=1,2,\dots,T$$

F_{it} denotes the trade flow in country i in the t th year. γ is a vector of the coefficients of explanatory variables. X_{1it} is a list variables measuring the economy, population, government effectiveness, access to the internet, performance of export sectors, and technological level of country i at time t ⁷³. The indicators used to represent these variables and their data sources are presented in Table A. 5. ρ_1 is the scalar coefficient. Based on the residual $\hat{\varepsilon}_{it}$ from the first-step estimation, this study considered the following model for the second-stage estimation of the 2SIV⁶⁸:

$$\ln SDG_{imt} = c_i + \rho_2 \ln SDG_{i,t-1} + \lambda_1 \sum_{j \neq i} w_{ijt}^1 \ln SDG_{jnt} + X_{2it} \beta + \delta \hat{\varepsilon}_{it} + v_{it}$$

SDG_{imt} and SDG_{jnt} are a pair of SDG indicators that were determined in Step 1, which respectively indicate the interaction receiver m and generator n . ρ_2 is the scale coefficient. λ_1 is the spatial coefficient which can be used to measure the transboundary SDG interactions under the transmission channel of international trade. w_{ijt}^1 is spatial a weight matrix related to international trade between country i and country j in year t . β is a vector of coefficient of the explanatory

variables. X_{2it} denotes the explanatory variables, which are shown in Table A. 6. The positive variables were transformed by taking their natural logarithms in the spatial dynamic panel data model. ”

Regarding the various flows, their directionality can be complex, as factors like ocean currents can change over time and space. This makes identifying their directions challenging. Consequently, we have followed prior studies to construct a spatial weight matrix that considers the magnitude of flow across regions, which represents connection strength²⁻⁴. We have acknowledged the challenge of identifying flow direction as an area for future research in lines 348-356 in Discussion section.

“Modelling flows such as ocean currents and wind patterns pose interesting methodological challenges given their multidirectional, changing dynamics over varying temporal and spatial scales. However, greater precision in characterising connectivity tendencies may considerably enhance our understanding of sustainability linkages. Future research should investigate novel approaches to systematically tracking variations in flow vectors—such as harnessing remote sensing data—and integrating this directional flow of information into spatial regression frameworks. This may entail simulating transport processes or calibrating networks via hydrodynamic or atmospheric modelling. Capturing the full complexity of flow regimes may provide unprecedented insights into the causal relationships among different countries.”

References:

1. Pham-Truffert, M., Metz, F., Fischer, M., Rueff, H. & Messerli, P. Interactions among Sustainable Development Goals: Knowledge for identifying multipliers and virtuous cycles. *Sustainable development* 28, 1236–1250 (2020).
2. You, W. & Lv, Z. Spillover effects of economic globalization on CO2 emissions: a spatial panel approach. *Energy economics* 73, 248–257 (2018).
3. Nan, S., Huo, Y., You, W. & Guo, Y. Globalization spatial spillover effects and carbon emissions: What is the role of economic complexity? *Energy Economics* 112, 106184 (2022).
4. Cheng, T., Wang, J., Haworth, J., Heydecker, B. & Chow, A. A dynamic spatial weight matrix and localized space–time autoregressive integrated moving average for network modeling. *Geographical Analysis* 46, 75–97 (2014).

5. I don't think the figures, e.g. figure 4, are very easy to read. Why are low-income countries in an outer circle? What do the circles mean, anyways? I don't think working with circles is very intuitive in this case and I would encourage the authors to rethink this.

Re: We appreciate your valuable feedback on this issue. The placement of low-income countries in either the outer or inner circle was purely a demonstration choice and carries no difference in meaning. However, recognizing the potential for confusion, we've updated the diagram with a more intuitive design to facilitate clearer understanding, as shown below.

Figure 4. Magnitude and components of transboundary SDG interactions by income group. Spatial interaction index by income groups (a) and share of the components of spatial interaction index by income groups (b). The sum of synergistic effects and trade-off effects equals the spatial interaction index. There are four income groups: high-, upper-middle, lower-middle, and low-income groups (Table A. 4). The pie charts utilize a dark blue color to represent the proportion of transboundary synergistic effects.

6. Crucially, SDG interactions between countries can result in mutual SDG achievement, if they are synergistic. However, what if one country does work “against” SDG achievement for some reason? Then, both countries – given the interactions – would mutually fail, right? This is different from non-synergistic interactions (where one countries’ action towards SDG would result in another countries action against an SDG). So synergies can also result in vicious circles if the “synergy” develops in “negative” ways, that is, against SDG achievement. Meaning that if country A fails, country B also fails...In the article related to the Pham-Truffert et al. database, there’s some discussion about this. This is crucial for the interpretation of results in this article as well, so the authors need to discuss this. It can change the interpretation of findings quiet a bit.

Re: That is an interesting point! We have included this new perspective in Discussion section and cited this useful paper.

“Addressing a single SDG indicator within one country can automatically strengthen not only the same indicator but other indicators across connected nations through transboundary synergistic effects. Conversely, if these transboundary synergies develop in negative ways, such as a country failing to progress on certain indicators or regressing, they may result in vicious cycles in which setbacks are multiplied and transmitted to other countries²¹. This highlights the risks and emphasizes the necessity to convert vicious inter-country cycles into virtuous ones. Systemic interlinkages form either virtuous or vicious cycles, indicating that transformations must be pursued intentionally to initiate desirable co-benefits and multiplication effects across borders. Pursuing progress in a coordinated manner across countries could set the stage for mutually reinforcing advances in the SDGs at a global scale.”

Reference:

21. Pham-Truffert, M., Metz, F., Fischer, M., Rueff, H. & Messerli, P. Interactions among Sustainable Development Goals: Knowledge for identifying multipliers and virtuous cycles. *Sustainable development* 28, 1236–1250 (2020).

7. - The text at the beginning of the results section is not straightforward – it is never clear to what total the percentages relate to. Please clarify.

Re: Many thanks for your comment. The total percentages represent the combined shares of both synergistic and trade-off linkages. We have revised the first paragraph to make it more straightforward in lines 93-105 in Result section.

*“Through the transmission channels of both international trade and nature-caused flows (incorporating river flow, ocean currents, and air flow), the transboundary synergistic linkages were more pronounced than their trade-off counterparts. Specifically, amongst the **transboundary linkages, which include synergistic and trade-off linkages**, 73.68% of the linkages resulting from international trade were synergistic (Figure 2a and b). Similarly, 81.82% of linkages originating from nature-caused flows were synergistic (Figure 2c and d). These results also highlight that, compared with interaction linkages resulting from nature-caused flows, linkages originating from international trade are generally more susceptible to counterproductive effects, potentially undermining joint efforts towards the SDGs. To provide further clarity, within the sphere of international trade, trade-off linkages accounted for a considerable 26.32% (calculated as 100%–73.68%) of the total SDG interaction linkages (Figure 2a and b). This percentage is notably higher than the 18.18% (calculated as 100%–81.82%) associated with nature-caused flows, as shown*

in Figure 2c and d.”

Figure 2. Transboundary synergistic and trade-off linkages across SDG indicators among countries. Transboundary synergistic (a) and trade-off (b) linkages across SDG indicators via international trade. Transboundary synergistic (c) and trade-off (d) linkages across SDG indicators through nature-caused flows. For better demonstration, the SDG indicators belonging to the same SDG target were grouped together based on the UN Global Indicator Framework for Sustainable Development Goals developed by the Inter-Agency and Expert Group on SDG Indicators (IAEG-SDGs)⁴⁷. The left and right axes respectively denote the SDG targets belong to SDG 1 to SDG 17, with the former serving as interaction generators and the latter as interaction receivers. The colour bars show the absolute value of transboundary interactions.

8. The discussion is quite thin. The general statement that “the analysis shows that countries depend on each other” is a natural results given the research design, not a specific finding. I would encourage the authors to be more specific in their discussion, and to improve the link of the discussion elements to the literature.

Re: Thanks a lot for your valuable comment. We have deleted this general statement, highlighted more specific findings, and included more elements related to previous studies in lines 272-283 in Discussion section.

“Some research credits globalisation and openness with benefiting sustainability and economic development by invoking Ricardo’s theory of comparative advantage^{4,5,35}. However, other scholars argue that openness and globalisation run contrary to sustainability goals, based on the pollution haven hypothesis^{5,36,37}. While theories of comparative advantage highlight the welfare gains of interconnectedness, the pollution haven hypothesis introduces an important caveat regarding potential cross-border regulatory distortions and their impact on sustainability outcomes in integrated economies at different development levels. The study demonstrated that transboundary synergistic effects through international trade were the dominant form of interactions across borders, revealing the overall positive impact of globalisation and openness on advancing global sustainability in an interconnected world. While trade-offs exist in some issues, the predominance of cross-border coordination benefits underscores how collective progress is enhanced through continued cooperation at a global scale. ”

References:

4. Taghvaei, V. M., Nodehi, M., Arani, A. A., Jafari, Y. & Shirazi, J. K. Sustainability spillover effects of social, environment and economy: mapping global sustainable development in a systematic analysis. *Asia-Pac J Reg Sci* 7, 329 – 353 (2023).
5. Mohamad Taghvaei, V., Assari Arani, A. & Agheli, L. Sustainable development spillover effects between North America and MENA: Analyzing the integrated sustainability perspective. *Environmental and Sustainability Indicators* 14, 100182 (2022).
35. Umar, M., Ji, X., Kirikkaleli, D., Shahbaz, M. & Zhou, X. Environmental cost of natural resources utilization and economic growth: can China shift some burden through globalization for sustainable development? *Sustainable Development* 28, 1678 – 1688 (2020).
36. Cole, M. A. & Elliott, R. J. R. Determining the trade – environment composition effect: the role of capital, labor and environmental regulations. *Journal of Environmental Economics and Management* 46, 363 – 383 (2003).
37. Cave, L. A. & Blomquist, G. C. Environmental policy in the European Union: Fostering the development of pollution havens? *Ecological Economics* 65, 253 – 261 (2008).

9. Related to this, I would like to see more discussion on how some findings are related, and potentially interpreted. E.g., are there any potential explanations on why high-income countries have stronger interactions with neighboring countries than with distant ones? Also, it’s not surprising that natural flows result in stronger relations between neighboring countries than with distant

countries (e.g. flows of water and air). Can the authors open up the discussion a bit more and discuss which types of natural flows might behave similarly, or differently, than those included in the empirics?

Re: Thank you for your insightful comment. We agree that it would be beneficial to delve deeper into the interpretation of our finding.

We've detailed the reasons behind the noticeably stronger interactions from high-income countries, as well as why they tend to engage more intensely with countries outside their geographic vicinity rather than their neighbors, in the Result section on lines 186-196 and lines 237-251.

“Compared to low, lower-middle, and upper-middle income groups, high income countries bear a greater responsibility for the influence of their domestic actions on the achievement of the 17 SDGs in other countries, as the magnitude of their transboundary SDG interactions accounted for the largest proportion of the total transboundary SDG interactions of the four income groups (sum of spatial interaction index), at 61.22% (Figure 4). High income countries demonstrated strong transboundary interactions with other countries; however, the population of high income countries over 2010–2020 accounted for an average of only 14.18% of the global population, based on data from the World Bank. Despite representing a relatively small fraction of the global population, high income countries are often characterised by robust economies, advanced technologies, and considerable political influence, which may amplify their roles in SDG interactions.”

“Interestingly, a common trend emerged among all income groups: they tended to establish more intense synergistic relationships with trade partners outside their immediate geographic vicinity than with neighbouring trade partners (Figure 5b). However, this trend was especially pronounced in high income countries, which demonstrated a notably stronger tendency towards synergistic effects with non-neighbouring trade partners (Figure 5b).....This can be attributed to the extensive global practices and international influence of high income countries. High income countries often have widespread networks of investments and trade relationships worldwide, facilitating stronger interactions with non-neighbouring countries. Participation in various international accords and organisations encourages these countries to extend their relationships beyond their immediate geographic sphere, fostering more intensive interactions globally. Moreover, their relatively advanced technological infrastructure enables efficient communication and transportation over long distances.”

We've incorporated a discussion about the similarities and differences in the behavior of natural flows compared to the empirics. You can find this in the Results section on lines 128-136.

“Figure 2 also revealed that a single SDG indicator can influence both its counterpart in other countries and various other SDG indicators. For example, air pollutants (SDG 11.6) in other countries, through air flow, can profoundly impact both ambient air quality in other countries and 14 other indicators across multiple SDGs in focal countries(Figure 2c and d). This ripple effect may influence health outcomes and economies—leading, for example, to a potential reduction in work time and productivity, which aligns with SDG 8 (promoting decent work and economic growth). Moreover, it can also impact biodiversity, as represented by SDG 15 (life on land), by modifying habitats and harming wildlife. This discovery underscores the complex and interconnected nature of SDG interactions mediated by natural flows such as air.”

10. Also, in the discussion and elsewhere, there is a strong emphasis on Tobler’s first law of geography. While not being a specialist in the field of geography, I would be highly surprised if that law had not been questioned elsewhere, and earlier, in other contexts. So I don’t really understand the strong focus of the authors on their results contradicting this law. I would put less emphasis on that. Plus the references to that law and the literature relying on it (and probably sometimes also contradicting it) are lacking.

Re: Thanks for this great idea. We have revised the Discussion by adding references related to that law and the literature relying on and against it in lines 293-312, as shown in the below paragraph. Our findings indicate that this law does not hold true in all circumstances in the real world. However, we recognize that these contradictions to Tobler's law are not universally applicable either. Therefore, we have been careful not to overly emphasize these contradictions throughout the manuscript.

“Tobler’s first law of geography—that everything is related to everything else, but near things are more related than distant ones—serves as a foundational principle in numerous research fields, including spatial analysis in epidemiology³⁸, crime pattern analysis³⁹, economic development^{40,41}, and environmental issues⁴². The continuing relevance and applicability of this law are evidenced by the broad range of methodologies and concepts that have been developed based on it. In modern times, as sustainability issues increasingly intersect with geographical considerations, there has been a increasing interest in revisiting and further exploring Tobler’s law, especially with respect to SDGs⁴³. While the law’s core principle remains valid in many instances, studies have illuminated the complex ways in which distance can

shape interrelations, sometimes counterintuitively¹⁶. For instance, some studies have highlighted habitat losses triggered by distant consumption through international trade^{44,45}. The negative effects of distant activities on sustainable fisheries further challenge this geographical principle^{46,47}. The results of this study suggested that the synergistic effects were 5.65% more pronounced in interactions between trade partners that did not share borders compared with those between neighbouring counterparts. Due to globalisation, different countries have become more connected and less geographically limited through international trade. Globally, non-neighbouring countries can benefit from comparative advantages by diversifying their traded goods and services, allowing them to interact more with each other than with neighbouring countries. While Tobler's law remains valuable for understanding geographical influences, recent research has revealed the importance of considering a broader array of factors, including non-proximate influences. ”

References:

38. Ulimwengu, J. & Kibonge, A. *Spatial spillover and COVID-19 spread in the U.S.* *BMC Public Health* 21, 1765 (2021).
39. Carboni, O. A. & Detotto, C. *The economic consequences of crime in Italy.* *Journal of Economic Studies* 43, 122–140 (2016).
40. Peng, W., Yin, Y., Kuang, C., Wen, Z. & Kuang, J. *Spatial spillover effect of green innovation on economic development quality in China: Evidence from a panel data of 270 prefecture-level and above cities.* *Sustainable Cities and Society* 69, 102863 (2021).
41. Marinos, T., Belegri-Roboli, A., Michaelides, P. G. & Konstantakis, K. N. *The spatial spillover effect of transport infrastructures in the Greek economy (2000–2013): A panel data analysis.* *Research in Transportation Economics* 94, 101179 (2022).
42. Nan, S., Huo, Y., You, W. & Guo, Y. *Globalization spatial spillover effects and carbon emissions: What is the role of economic complexity?* *Energy Economics* 112, 106184 (2022).
43. Manning, N., Li, Y. & Liu, J. *Broader applicability of the metacoupling framework than Tobler's first law of geography for global sustainability: a systematic review.* *Geography and Sustainability* (2022).
44. Green, J. M. et al. *Linking global drivers of agricultural trade to on-the-ground impacts on biodiversity.* *Proceedings of the National Academy of Sciences* 116, 23202–23208 (2019).
45. Lenzen, M. et al. *International trade drives biodiversity threats in developing nations.* *Nature* 486, 109–112 (2012).
46. Carlson, A. K., Taylor, W. W., Rubenstein, D. I., Levin, S. A. & Liu, J. *Global marine fishing across space and time.* *Sustainability* 12, 4714 (2020).

47. Carlson, A. K., Taylor, W. W. & Liu, J. Using the telecoupling framework to improve Great Lakes fisheries sustainability. *Aquatic ecosystem health & management* 22, 342–354 (2019).

11. I think the different methodological steps in the methods part would strongly benefit from an example so that the reader can follow the calculations through a specific example. Ideally the same example is used for all subsequent steps.

Re: This is a very useful suggestion. Thank you very much. We have included a specific example to illustrate the whole process in obtaining the spatial interaction index in lines 516-529 in Method section.

“Using SDG 7.1.1 (access to electricity) as an example, this study analysed the magnitude and direction of impacts on a country’s performance in achieving SDG 7.1.1 from progress on SDG indicators in other countries. In Step 1, it was identified that the achievement of SDG 7.1.1 could be influenced by SDG 6.4.1 (water use efficiency) and SDG 7.3.1 (energy intensity). Spatial weight matrices were constructed to represent connections between countries through river basins and trade networks. Subsequently, this study row-standardised the weight matrices. This normalisation process prepared the data for spatial econometric modelling. The last step involved using spatial econometric models to calculate the spatial coefficients. These coefficients (both λ_1 and λ_2) indicate the direction and magnitude of transboundary interactions on SDG 7.1.1 outcomes in the focal country, respectively. If both coefficients were statistically significant at least 10%, the study would sum their absolute values and standardised this total into a single index from 0 to 100. This example shows how the performance of SDG 7.1.1 in focal countries could be influenced by other countries’ progress on SDG 6.4.1, through shared river flows powering hydropower, and SDG 7.3.1, through energy used in internationally traded goods and services.”

REVIEWER COMMENTS

Reviewer #1 (Remarks to the Author):

The revised manuscript has well considered the comments, and improved substantially. However, it has some minor issues. For example, the sentence of the introduction section defines the three pillars of the sustainability as a subset of SDGs, while it is reversed. Instead, the three pillars of sustainability encompass the SDGs, not the SDGs encompass them. For this reason, the second sentence should find an active voice rather than a passive one. In addition, the third sentence of the introduction write "as discussed in the literature", before starting the literature. This statement is redundant.

The result section writes SDGs with 2 digits. For example, SDG 6.6, whereas SDGs have only one digit. The manuscript can recheck if it means "Target" or "SDG" by "SDG 6.6". Perhaps, "Target" has two digits, instead of SDGs.

Finally, the abstract needs further editing. For instance, it lacks the research implication. Although the last sentence tried to play this role, but it is not successful. An alternative implication of this research can be about promoting globalization, international unions ,and world organizations.

Reviewer #2 (Remarks to the Author):

Dear authors,

Thank you very much for responding to my report.

In the revised version of the manuscript, you discuss spatial models and the potential endogeneity of the weights matrix. You also describe a two-stage instrumental variable (2SIV) estimation of the spatial econometric model. However, I could not find the results and a corresponding discussion of the analysis. The only statement I could find was on lines 535-539: "The last step involved using spatial econometric models to calculate the spatial coefficients. These coefficients (both λ_1 and λ_2) indicate the direction and magnitude of transboundary interactions on SDG 7.1.1 outcomes in the focal country, respectively. If both coefficients were statistically significant at least 10%, the study would sum their absolute values and standardised this total into a single index from 0 to 100." Did you actually perform the estimation? I could not find these results.

Reviewer #3 (Remarks to the Author):

This is the second time I review this manuscript. I'm very happy with how the authors took my previous comments into account and how they explained and justified their choices related to the changes. Overall, I'm happy to recommend publication.

Point-to-Point Response to Reviewer's Comments

Reviewer #1

1. The revised manuscript has well considered the comments, and improved substantially. However, it has some minor issues. For example, the sentence of the introduction section defines the three pillars of the sustainability as a subset of SDGs, while it is reversed. Instead, the three pillars of sustainability encompass the SDGs, not the SDGs encompass them. For this reason, the second sentence should find an active voice rather than a passive one.

Re: Thanks a lot for your kind and valuable comments. We have revised this sentence as follows:

“The three main pillars of sustainability—economy, society, and environment—encompassed these goals.”

2. In addition, the third sentence of the introduction write “as discussed in the literature”, before starting the literature. This statement is redundant.

Re: Thank you for your suggestion. We have deleted this statement as follows:

The original sentence is: *“Sustainability, as discussed in the literature, is often approached from two perspectives: weak and strong sustainability”*

The revised sentence is: *“Sustainability is often approached from two perspectives: weak and strong sustainability”*

3. The result section writes SDGs with 2 digits. For example, SDG 6.6, whereas SDGs have only one digit. The manuscript can recheck if it means “Target” or “SDG” by “SDG 6.6”. Perhaps, “Target” has two digits, instead of SDGs.

Re: Thank you for your insightful suggestion. In our context, we actually meant 'target' rather than 'SDG', so we have revised 'SDG 6.6' to 'target 6.6'. We have also thoroughly reviewed the manuscript and made similar revisions where necessary. We appreciate your valuable feedback.

“In the channel of nature-caused flows, the SDG indicator that affected the most SDG indicators in other countries was related to target 6.6. (protect and restore water-related ecosystems) (Figure 2c and d). Attempts to improve the performance of target 6.6 in interconnected countries may interact with the performance of 18 SDG indicators in focal countries (Figure 2c and d).”

4. Finally, the abstract needs further editing. For instance, it lacks the research implication. Although the last sentence tried to play this role, but it is not successful. An alternative

implication of this research can be about promoting globalization, international unions ,and world organizations.

Re: This is a very useful suggestion. We have followed your instructions to revise the last sentence in the abstract to highlight the research implication, as shown below:

“These findings highlight the significance of collaborative solutions among countries by utilizing international institutions as platforms to promote globalisation. These collective efforts allow nations to leverage transboundary synergies and facilitate the achievement of SDGs worldwide”

Reviewer #2

1. In the revised version of the manuscript, you discuss spatial models and the potential endogeneity of the weights matrix. You also describe a two-stage instrumental variable (2SIV) estimation of the spatial econometric model. However, I could not find the results and a corresponding discussion of the analysis. The only statement I could find was on lines 535-539: "The last step involved using spatial econometric models to calculate the spatial coefficients. These coefficients (both λ_1 and λ_2) indicate the direction and magnitude of transboundary interactions on SDG 7.1.1 outcomes in the focal country, respectively. If both coefficients were statistically significant at least 10%, the study would sum their absolute values and standardised this total into a single index from 0 to 100." Did you actually perform the estimation? I could not find these results.

Re: Thank you for your valuable reminder and insightful suggestion. Given the extensive nature of our research, involving 768 regression models based on the spatial econometric method to explore transboundary interactions between SDG indicators, space restrictions within the main manuscript limit the presentation of all empirical results. As you rightly pointed out, the empirical results may not be easily found. Therefore, we've respectively selected and demonstrated the empirical results of the SDG targets that exhibit the most intensive linkages via the transmission channels of international trade and naturally-caused flows. These are displayed in Table 1. Then, we've added some discussion to better describe these findings in lines (124-153) in the Results section.

Additionally, to more effectively illustrate all the spatial coefficients, which represent transboundary interactions derived from the 768 of the spatial econometric models, we have included some explanation in Figure 2. This note emphasizes that all spatial coefficients with a significance level of at least 10% are represented on the color bars.

*"Table 1. Empirical results of transboundary interactions based on a two-stage instrumental variable (2SIV) of the spatial econometric model. The indicators chosen to represent SDG targets 7.1, 6.4, 1.4, and 6.6 are "Proportion of population with primary reliance on clean fuels and technology", "Water Use Efficiency", "Proportion of population using basic sanitation services", and "Lakes and rivers seasonal water area (% of total land area)" respectively. Detailed information regarding the additional variables and the rationale behind their selection can be found in Tables A.5 and A.6. Standard errors are provided in parentheses. Significance at the 1%, 5% and 10% levels is denoted by ***, ** and *, respectively.*

First stage estimation	Explained variable: target 6.4		Explained variable: target 1.4		
GDP	0.091*** (0.033)	Spatial lag of target 7.1 (trade flow)	0.189*** (0.043)	Spatial lag of target 6.6 (trade flow)	0.038*** (0.013)
Population	0.071** (0.035)	Time lag	0.234*** (0.079)	Spatial lag of target 6.6 (river flow)	0.094*** (0.013)
Government	0.034	Economy	0.020	Time lag	-0.003

	(0.030)		(0.026)		(0.059)
Internet	0.028 (0.034)	Education	-0.029 (0.028)	Economy	-0.001 (0.007)
Export value	0.102*** (0.035)	Technology	0.053*** (0.024)	Environment	0.017 (0.010)
Technology	0.038 (0.027)	Governance	0.045 (0.028)	Education	0.019* (0.010)
		Agriculture	-0.006 (0.025)	Governance	0.018** (0.009)
		Residual from the first stage	0.039* (0.023)	Population	-0.002 (0.009)
				Residual from the first stage	-0.018** (0.007)

“In the international trade channel, indicators related to target 7.1 (to ensure universal access to affordable, reliable, and modern energy services) had the most (26) linkages with the SDG indicators in other countries (Figure 2a and b). These indicators were linked to various basic human needs and the environment in other countries, such as basic drinking water and sanitation services (four linkages with target 1.4), agricultural productivity (two linkages with target 2.3), water-use efficiency (two linkages with target 6.4), housing (three linkages with target 11.1), and biodiversity (two linkages with target 15.5) (Figure 2a and b). For instance, via the channel of international trade, the spatial lag term of target 7.1 proved to be both significant and positive (**Table 1**). This implies that the achievement of target 6.4 in certain countries could be promoted by synergistic effects stemming from the progress their trade partners have made towards target 7.1. This extensive network of linkages may be primarily attributed to the fundamental role of energy in many sectors. The production, distribution, and consumption of energy through international trade can have far-reaching transboundary impacts on various aspects of society and the environment.

In the channel of nature-caused flows, the SDG indicator that affected the most SDG indicators in other countries was related to target 6.6. (protect and restore water-related ecosystems) (Figure 2c and d). Attempts to improve the performance of target 6.6 in interconnected countries may interact with the performance of 18 SDG indicators in focal countries (Figure 2c and d). For example, the spatial lag of term of target 6.6 (river flow) is 0.094, with statistical significance at the 1% level (**Table 1**), suggesting the progress of SDG 1.4 of some countries can be promoted by the other countries’ actions towards achieving target 6.6 through the transboundary rivers. The actions of other countries focused on protecting and restoring water-related ecosystems may have transboundary SDG impacts, thereby creating numerous benefits for focal countries.”

Figure 2. Transboundary synergistic and trade-off linkages across SDG indicators among countries. Transboundary synergistic (a) and trade-off (b) linkages across SDG indicators via international trade. Transboundary synergistic (c) and trade-off (d) linkages across SDG indicators through nature-caused flows. For better demonstration, the SDG indicators belonging to the same SDG target were grouped together based on the UN Global Indicator Framework for Sustainable Development Goals developed by the Inter-Agency and Expert Group on SDG Indicators (IAEG-SDGs)³⁴. The left and right axes respectively denote the SDG targets belong to SDG 1 to SDG 17, with the former serving as interaction generators and the latter as interaction receivers. The colour bars show the absolute values of spatial coefficients that are statistically significant, which serve to represent the magnitude of transboundary interactions. These values were derived from 768 regression models based on spatial econometric methods, elaborated further in the Methods section under “Step 4: Quantify transboundary SDG interactions”. A darker colour, corresponding to a higher absolute value of the spatial coefficient, signifies stronger transboundary interactions.

Reviewer #3

1. This is the second time I review this manuscript. I'm very happy with how the authors took my previous comments into account and how they explained and justified their choices related to the changes. Overall, I'm happy to recommend publication.

Re: Thank you very much for your valuable time and constructive comments in the last round of revision, which have significantly improved our manuscript.

REVIEWERS' COMMENTS

Reviewer #2 (Remarks to the Author):

Thank you very much for addressing and clarifying my outstanding issues. The only thing left for me to do is to congratulate you on a fine paper. I hope it gets attention.